# Task-dependent representations of stimulus and choice in mouse parietal cortex

Gerald N. Pho [1,2,3], Michael J. Goard [1,2,4,5], Jonathan Woodson[1,2], Benjamin Crawford[1,2] & Mriganka Sur[1,2]

The posterior parietal cortex (PPC) has been implicated in perceptual decisions, but whether its role is specific to sensory processing or sensorimotor transformation is not well understood. Here, we trained mice to perform a go/no-go visual discrimination task and imaged the activity of neurons in primary visual cortex (V1) and PPC during engaged behavior and passive viewing. Unlike V1 neurons, which respond robustly to stimuli in both conditions, most PPC neurons respond exclusively during task engagement. To test whether signals in PPC primarily encoded the stimulus or the animal's impending choice, we image the same neurons before and after re-training mice with a reversed sensorimotor contingency. Unlike V1 neurons, most PPC neurons reflect the animal's choice of the new target stimulus after re-training. Mouse PPC is therefore strongly task-dependent, reflects choice more than stimulus, and may play a role in the transformation of visual inputs into motor commands.

[1] Department of Brain and Cognitive Sciences, Massachusetts Institute of Technology, Cambridge, MA 02139, USA. [2] Picower Institute for Learning and Memory, Massachusetts Institute of Technology, Cambridge, MA 02139, USA. [3] Department of Organismic and Evolutionary Biology, Harvard University, Cambridge, MA 02138, USA. [4] Department of Molecular, Cellular, and Developmental Biology, University of California, Santa Barbara, Santa Barbara, CA 93106, USA. [5] Department of Psychological & Brain Sciences, University of California, Santa Barbara, Santa Barbara, CA 93106, USA. These authors contributed equally: Gerald N. Pho, Michael J. Goard. Correspondence and requests for materials should be addressed to M.S. (email: msur@mit.edu)

Perceptual decision-making involves multiple cognitive processes, including processing of sensory stimuli, accumulation of evidence, and transformation of sensory information into an appropriate motor plan. Although many brain regions have been implicated in perceptual decisions, dissociating their individual contribution to these different processes remains a challenge.

In particular, the posterior parietal cortex (PPC) has been hypothesized to play a key role in at least some types of decision tasks in both primates[1,2] and in rodents[3–5]. However, the specific role of rodent parietal cortex, and whether its function is homologous to that of primates, remains unclear. While rodent PPC plays a minimal role in simple auditory[6] and whisker-based[7] decision tasks (but see[8]), multiple groups have demonstrated that it is causally necessary for make decisions on the basis of visual stimuli[3,4,9,10].

However, the specific role of PPC in visual decision-making remains unclear. Some argue that rodent PPC may ultimately be more homologous to extrastriate cortex in processing sensory signals that are accumulated elsewhere for decision-making[10]. Indeed, both anatomical projection studies[11,12], as well as functional mapping studies[13,14] indicate that PPC may overlap with or contain a group of retinotopically-organized extrastriate areas that are rostral to primary visual cortex (V1). Others suggest that PPC reflects internal biases related to the value of past stimuli[8] or actions[9]. However, removal of internal biases cannot fully explain the decision-making deficits induced by inactivation of PPC during visual decision tasks[3,4,10]. A third, alternative possibility is that PPC may play a role in the mapping of visual stimuli to motor commands. If this were the case, one may expect that activity in PPC would be highly task-dependent, reflecting the animal's decision depending on learned sensorimotor contingencies.

Here, we use population calcium imaging to investigate these possibilities by measuring activity in PPC and in V1 during a go/no-go lick-based visual discrimination task. Having previously demonstrated the necessity of PPC in the performance of this task[3], we seek in this work to investigate its specific role in either sensory processing or sensorimotor transformation by manipulating task engagement and learned task contingencies. V1 neurons exhibit robust visual responses during both task engagement and passive viewing of stimuli that remain stable after task contingency reversal. By contrast, PPC responses are largely specific to task performance and reflect the animal's choice before and after task contingency reversal. Our results are consistent with a role of the mouse posterior parietal cortex in transforming visual information to motor commands during perceptual decisions.

## Results

**Imaging V1 and PPC during task performance and passive viewing**. We trained mice on a head-fixed lick/no-lick visual discrimination task (Fig. 1a, b), similar to previous designs[3,15,16]. Water-restricted mice discriminated between a target stimulus (horizontal grating drifting upwards, 0° from vertical, Stimulus A) which was rewarded with water, and a non-target stimulus of an orthogonal orientation (vertical drifting upwards rightwards, 90°, Stimulus B). Lick responses to the non-target grating were discouraged by punishment with a small, aversive drop of quinine. Quinine concentrations were chosen to discourage licking to non-targets, but without inducing long-lasting changes in motivation on subsequent trials (Supplementary Fig. 1). A retractable lick spout was presented immediately after stimulus presentation (2 s) and retracted on every trial. This restricted the animal's lick response to the "response" period (1.5 s) and allowed us to

separately assess perception and action (Fig. 1b). Video recording of the mice during the stimulus period confirmed that mice withheld licking until the spout became available during the response epoch[3]. Mice ($n = 15$) achieved high levels of performance on the task (Fig. 1c; d-prime or $d'$, $2.23 \pm 0.18$; mean ± SEM), with a bias towards licking, resulting in more false alarms than misses.

We have previously demonstrated using a version of this task with a 4 s delay period that inactivation of either V1 or PPC during the stimulus period disrupts behavioral performance, whereas inactivation during later periods has no effect[3]. Stimulus-period activity in PPC is therefore necessary for the task, but it is unclear whether such activity is important for sensory processing, attentional engagement, decision formation, or motor planning. To help distinguish between these possibilities, we measured neural activity in PPC and V1 under two different conditions: engagement in the behavioral task as well as passive viewing of the same visual stimuli (Fig. 1b). While "task engagement" may not be able to fully separate various non-sensory signals, we reasoned that this approach could allow us to identify pure sensory responses that were robust during both passive and engaged conditions.

We used two-photon microscopy and a volumetric imaging approach to image hundreds of neurons simultaneously in either V1 or PPC (see "Methods" section). After behavioral training, we injected AAV2/1 syn-GCaMP6s[17] into the two areas under stereotaxic guidance. We used a resonant scanning system combined with a z-piezo to concurrently record activity from several hundreds of GCaMP6-infected cells within layer 2/3 in a volume comprising four planes ($850 \times 850\,\mu m$) $20\,\mu m$ apart in depth, at an overall stack rate of 5 Hz. Images were corrected for $X–Y$ movement, and fluorescence traces were extracted from semi-automatically generated ROIs based on a pixel-wise activity map (see "Methods" section).

To investigate the effects of behavioral performance on neural responses, we imaged the same neurons in alternating blocks (5–10 min, 40–80 trials) of engaged behavior and passive viewing (Fig. 1b). During "engaged" behavior trials, the spout was extended on each trial during the response period. During "passive" trials, the spout was not extended during the response period but remained withdrawn and inaccessible from the animal. To avoid extraneous stimulus confounds, no additional cue was provided to signal engaged or passive blocks. Nonetheless, mice rapidly became aware after the first 1–2 trials of a block whether the spout would be available for a behavioral response, as confirmed in video recordings by the complete lack of licking during passive blocks. Discrimination performance was similarly high for both the first and subsequent engaged blocks.

We imaged from an average of 606 cells (range: 257–1057) in each population within either V1 or PPC, of which an average 135 cells (range: 21–364) exhibited significant task-related responses. Pilot experiments using transgenic mice with tdTomato expressed in PV + and SOM + interneurons revealed that calcium signals from these neurons were generally too weak to be measured with volume scanning, and therefore the vast majority of task-responsive cells were likely to be excitatory pyramidal neurons[3]. We first analyzed only the neural responses measured during correct Engaged trials (ignoring error trials), or during Passive viewing. In V1, many neurons showed a robust and reliable response to either the target or non-target stimulus, though some were modulated by engagement in the task. For example, one target-preferring cell showed a reliable passive response to the target stimulus that was moderately enhanced during task performance (Fig. 1d, left). A non-target preferring cell, however, exhibited a suppressed response during engagement in the task (Fig. 1d, right). By contrast, PPC neurons exhibited

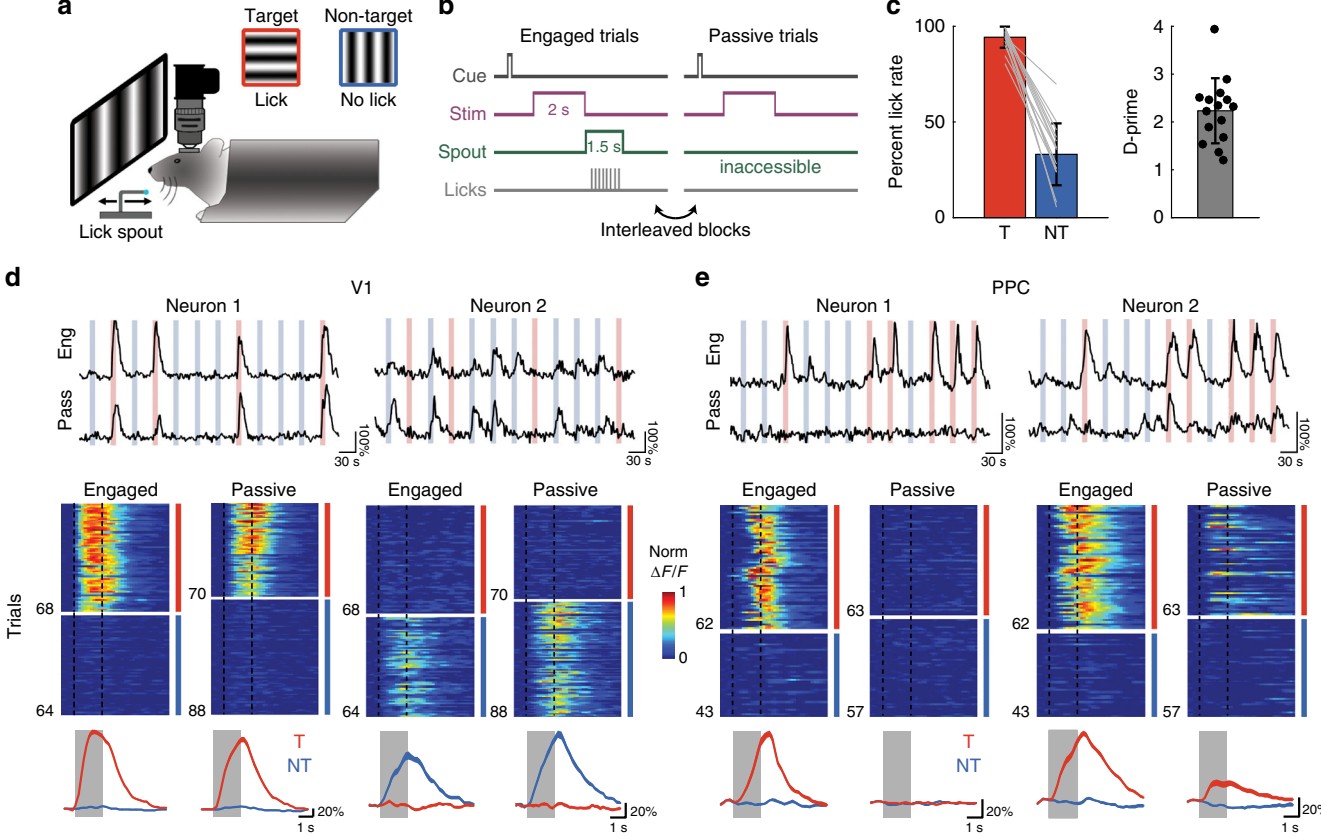

**Fig 1** Imaging calcium responses in V1 and PPC during engaged task performance and passive viewing. **a** Head-fixed mice were trained to perform a go, no-go lick-based visual discrimination task. A retractable lick spout was used to restrict lick responses to a specific epoch of the task. Licks following a target stimulus (red, horizontal drifting upwards, Stimulus A) were rewarded with water, while licks to non-target stimulus (blue, vertical drifting rightwards, Stimulus B) were punished with quinine. **b** Trial structure for Engaged and Passive conditions. After a brief auditory preparatory cue, a drifting grating was presented for 2 s. During Engaged trials, the retractable lick spout was presented immediately after stimulus offset for a minimum of 1.5 s. During Passive trials, the spout was inaccessible. Engaged and Passive trials were presented in blocks, which were usually interleaved. **c** Rate of licking on target (T, red) and non-target (NT, blue) for each mouse used in imaging experiments ($n = 15$). Behavioral performance was quantified as d-prime (mean across mice: 2.22). Error bars in this and all subsequent figures depict mean ± SEM. **d** Stimulus-evoked response of two V1 neurons, one target-selective (left column), and one non-target selective (right). Top, raw calcium response to multiple presentations of target (red) and non-target (blue) stimuli during both Engaged and Passive conditions. Middle, heatmap of trial-to-trial responses to target and non-target stimuli, presented in alternating blocks of Engaged (left) and Passive (right) trials, normalized to max response. Light gray shaded regions in this and subsequent figures demarcates duration of stimulus. Bottom, overlay of trial-averaged responses for each stimulus during Engaged (left) and Passive (right) conditions. Line thickness indicates mean ± SEM. **e** Same as **d** but for two PPC neurons

much stronger responses during task engagement compared to passive viewing, specifically for target stimuli. For example, one target-preferring cell had robust activity only during engagement (Fig. 1e, left), whereas another cell showed relatively weak passive responses that became stronger during task performance (Fig. 1e, right).

**Responses are visually-driven in V1 but task-gated in PPC.** To compare the overall response properties of V1 and PPC, we analyzed the responses of all neurons that had significant stimulus-period activity during the task. We focused on stimulus-period responses given that inactivation during this period disrupts behavioral performance[3]. A total of 1915 neurons (18% of all neurons, 18 fields, 9 mice) in V1 (Fig. 2a, b) and 3524 neurons (26% of all neurons, 22 fields, 10 mice) in PPC (Fig. 2c, d) were significantly responsive during the task. Two striking differences between V1 and PPC were immediately apparent when examining the trial-averaged responses. First, while V1 was about evenly split between target- and nontarget-preferring cells (60.6% ± 5.8%

target-preferring, Fig. 2e), there was a stronger bias in PPC towards the target stimulus (87.6% ± 7.4% target-preferring; V1 vs. PPC, $p = 4.68 \times 10^{-5}$, Wilcoxon rank-sum test). Secondly, while most task-responsive V1 neurons also responded during passive viewing (88.3% ± 6.8% with significant passive response, Fig. 2f), the majority of PPC neurons had responses that were gated by task engagement (only 29.7% ± 10.3% with significant passive response; V1 vs. PPC, $p = 4.71 \times 10^{-7}$, Wilcoxon rank-sum test).

The dramatic effect of task engagement on PPC responses could also be observed by comparing the selectivity of responses during Passive and Engaged conditions (Fig. 2g). We quantified selectivity using an ROC-based index that ranged from −1 to 1, with positive values indicating preference for the target stimulus (see "Methods" section for details). V1 had a large proportion of significantly selective ($p < 0.05$, permutation test) neurons in Passive conditions (85.3% ± 3.8% of cells), with a small but significant increase in Engaged conditions (91.8% ± 2.5% of cells; Passive vs. Engaged, $p = 4.97 \times 10^{-3}$, Wilcoxon signed-rank test). By contrast, PPC neurons were largely unselective during Passive

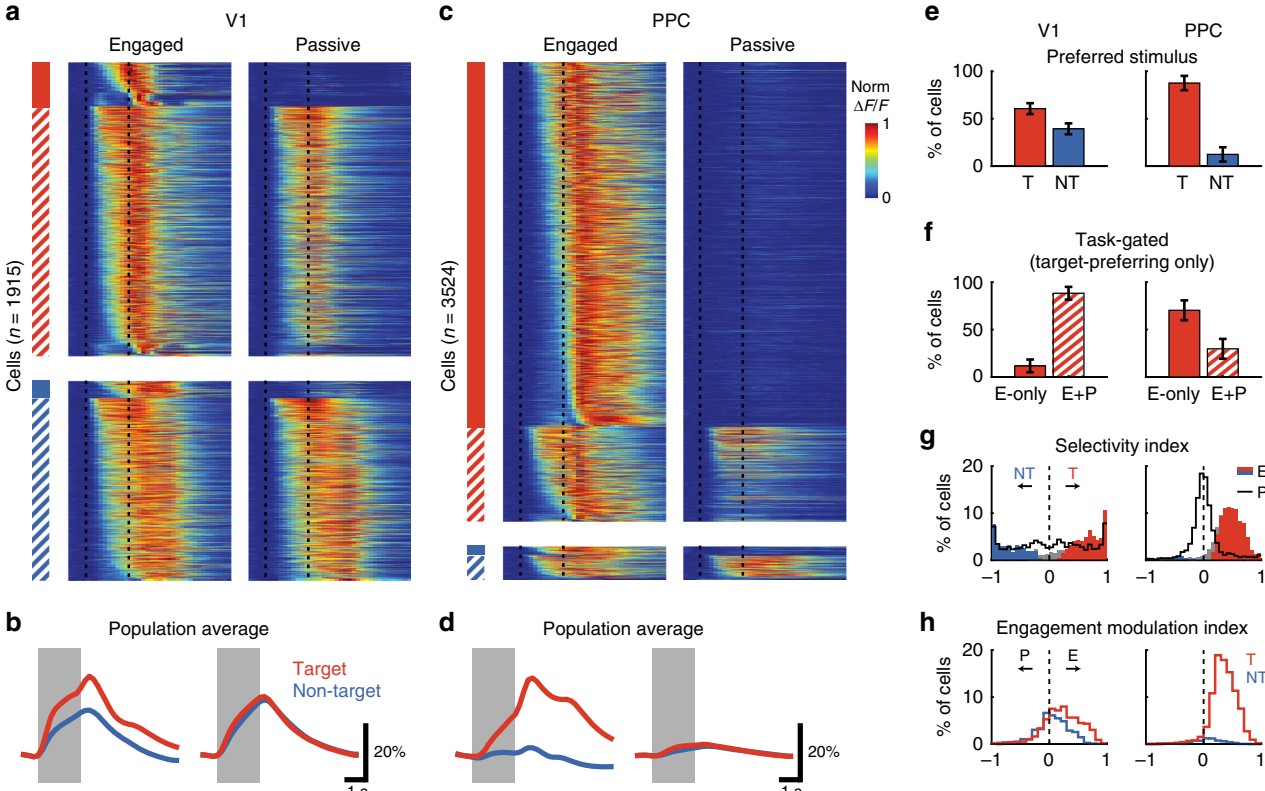

**Fig 2** V1 neurons respond passively, whereas PPC responses are gated by task engagement. **a** Trial-averaged responses of all task-responsive V1 neurons ($n = 1915$). Heatmap of all trial-averaged responses (preferred stimulus only) in both Engaged (left) and Passive (right) conditions, normalized by peak response. Neurons are separated into target-preferring (red) and non-target preferring (blue), and then into passive-responding (solid) and task-gated (hatched). Vertical dashed lines demarcate duration of stimulus. **b** Bottom, average response across V1 neurons to the target (red) and non-target (blue) in Engaged (left) and Passive (right) conditions. Line thickness indicates mean ± SEM. **c, d** Same as (**a, b**), but for PPC neurons ($n = 3524$). **e** Percentage of neurons in V1 (left) and PPC (right) that prefer target (T) or non-target (NT) stimulus. Error bars represent bootstrapped SEM across sessions. **f** Percentage of target-preferring neurons in V1 (left) and PPC (right) that are task-gated, i.e. that respond only during engagement (E-only), or that respond during both engaged and passive conditions (E + P). **g** Histogram of stimulus selectivity index for V1 (left) and PPC (right) during Engaged (filled bars) and Passive conditions (gray line). Positive selectivity indicates preference for Target. Colored bars indicate neurons with significant individual selectivity during Engaged trials. **h** Histogram of engagement modulation index for V1 (left) and PPC (right) for target- (red) and nontarget-preferring (blue) neurons

conditions (41.9% ± 10.8% selective), and yet became significantly selective to target trials during task engagement (89.5% ± 4.7% of cells; Passive vs. Engaged, $p = 5.95 \times 10^{-5}$, Wilcoxon signed-rank test). We also quantified the change in responses for engagement vs. passive viewing for each neuron using an engagement modulation index (Fig. 2h), which also ranged from −1 to 1, with positive values indicating increases with engagement. The mean modulation index was significantly above zero for target-preferring neurons in both V1 (0.227 ± 0.009, $n = 1148$; $p < 5.0 \times 10^{-4}$, permutation test, 2000 permutations) and PPC (0.351 ± 0.004, $n = 3292$; $p < 5.0 \times 10^{-4}$). Nontarget-preferring neurons in V1 showed weaker modulation (modulation index: 0.059 ± 0.010 for non-target, $n = 767$) than target-preferring neurons ($p = 9.07 \times 10^{-4}$, clustered Wilcoxon rank-sum test), indicating that the effect of engagement was stimulus-specific.

Comparing responses during engaged behavior and passive viewing therefore revealed different response properties in V1 and PPC. In V1, task engagement did not merely increase overall responsiveness (as would be expected with arousal), but instead modulated firing rates to enhance the contrast between target and non-target stimuli. By contrast, PPC responses were strongly target-selective and gated by behavior (Fig. 2). Furthermore, because this activity is selective to target trials in which the animal licks, PPC responses likely signal some aspect of the animal's choice (decision formation or motor planning), instead of simply

reflecting overall increases in arousal or attention during task engagement. A subset (~30%) of PPC neurons, however, do have significant passive responses, and could play a role in sensory processing.

**Error trials reveal sensitivity of PPC to both stimulus and choice.** How should one interpret this target-selective task engagement signal in PPC? One possibility is that PPC encodes a stimulus-specific signal that is perhaps boosted by task engagement. An alternative is that PPC activity reflects movement or action planning, since the animal licks on Engaged target (Hit) trials but not on Passive trials or Engaged nontarget (CR) trials.

To help disambiguate these possibilities, we analyzed the responses of V1 and PPC neurons on error trials. We compared activity in different behavioral or sensory conditions using single-neuron ROC analysis to isolate stimulus-specific signals from those related to the choice (Supplementary Table 1). Due to the difficulty in distinguishing between motor preparation and decision-related signals with our go/no-go paradigm, we define "choice" selectivity broadly as any premotor signal related to behavioral output independent of the stimulus. We focused on target-preferring neurons, as these represented the vast majority of responses in PPC, and restricted analysis to those imaging fields and behavioral sessions in which the mouse committed at

least five False Alarm (FA) trials (V1, $n = 1053$ cells from 16 fields; PPC, $n = 3034$ cells from 21 fields).

For V1, individual target-preferring neurons varied in their responsiveness on nontarget trials (Fig. 3a, left), but the average strength of the nontarget response was similar on FA trials compared to CR trials (Fig. 3b, left). By contrast, PPC neurons invariably showed stronger responses on FA vs. CR trials (Fig. 3a, b, right), indicating the presence of choice-related signals. This modulation was strongest during the response period, though still apparent in the stimulus period before any licks or punishment occurred. We quantified the selectivity of each neuron's stimulus-period response using an ROC-based index to compare trial types (Fig. 3c). Target-preferring V1 neurons exhibited strong selectivity for Hit compared to either CR or FA trials (Hit vs. FA, 0.605 ± 0.009, $p = 1.3 \times 10^{-4}$, clustered Wilcoxon signed-rank test), but weak modulation for FA vs. CR trials (0.014 ± 0.006, $p = 0.462$). On the other hand, PPC neurons showed significant selectivity for FA vs. CR trials (0.109 ± 0.003, $p = 0.014$), as well as selectivity for Hit compared to FA trials (0.348 ± 0.004, $p = 6.1 \times 10^{-5}$).

We also performed a time-dependent ROC analysis[3,18,19] to examine the dynamics of stimulus- and choice-related signals over the course of a trial. We quantified both the average selectivity (Fig. 3d) as well as the fraction of selective neurons (Fig. 3e) for each area as a function of time. While both areas exhibited significant stimulus-related selectivity (Hit vs. FA)

shortly after stimulus onset ($p < 0.05$ starting at $t = 0.2$ s for both areas, clustered Wilcoxon signed-rank test), only PPC exhibited any significant choice-related selectivity (FA vs. CR) in the stimulus period. The onset of choice selectivity was delayed ($p < 0.05$ starting at $t = 0.8$ s) compared to the stimulus signals and peaked during the response period. By contrast, choice-related selectivity in V1 did not become significant until the response period ($p < 0.05$ starting at 2.4 s).

These analyses suggest that PPC, unlike V1, is sensitive to both the stimulus and the impending choice of the animal. However, the comparison of Hit and FA trials to compute stimulus selectivity may be confounded by overt differences in motor output (see lick rate in Fig. 3b) or by covert differences in motor preparation. Could PPC activity be entirely explained by motor or choice-related signals?

While we cannot rule out covert preparatory effects, in a subset of sessions we eliminated differences in motor output by selecting Hit and FA trials with the same number of licks (Supplementary Fig. 2a, b). PPC responses in these sessions ($n = 846$ cells across 3 sessions) still exhibited stimulus selectivity (0.229 ± 0.004), as well as some motor-related sensitivity (0.130 ± 0.009) when comparing FA trials with different numbers of licks (Supplementary Fig. 2c–e). We also analyzed the responses of neurons on Miss trials, in which there is no motor output and presumably no motor preparation, using the subset of sessions with sufficient

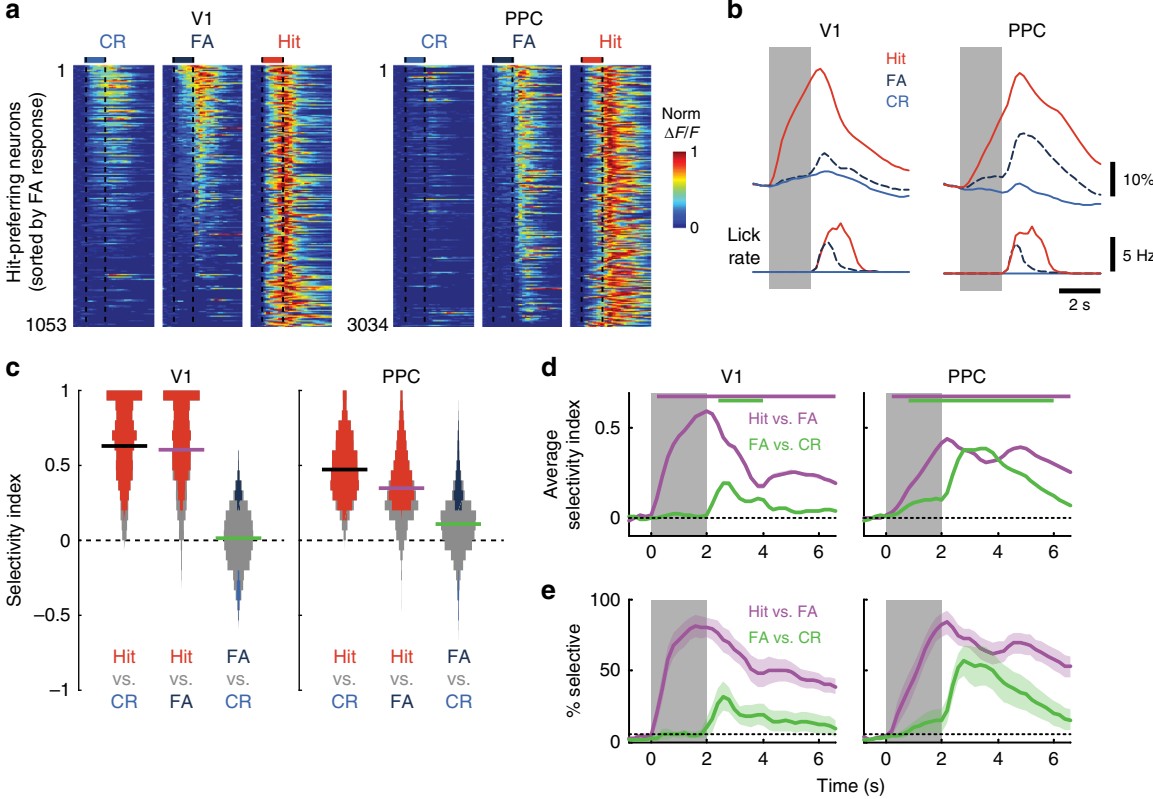

**Fig 3** Error trials reveal sensitivity of PPC to both stimulus and choice. Only Hit-preferring neurons were used in all panels of this figure. **a** Trial-averaged responses during False Alarm (FA) trials, for recordings with at least five FA trials, in V1 (left; 1053 neurons across 16 sessions) and in PPC (right; 3034 neurons across 21 sessions). Responses for each neuron are normalized by peak response across Correct Reject (CR, left, blue), FA (center, dark blue), and Hit trials (right, red). Neurons are sorted by descending strength of FA response. Vertical dashed lines demarcate duration of stimulus. **b** Population trial-averaged responses (top) during Hit, FA (dashed), and CR trials, for V1 (left) and PPC (right). Average lick rate (bottom). **c** Violin plots (rotated histograms) of selectivity index for V1 (left) and PPC (right) neurons, computed using stimulus-period responses. Three comparisons were made for each neuron in each area (Hit > CR, Hit > FA, FA > CR, see Supplementary Table 1). Horizontal lines indicate mean, and colored bars indicate neurons with significant individual modulation ($p < 0.05$). **d** Average selectivity between trial types, computed separately at each time-point. Colored bars at top indicate time points with significant non-zero selectivity ($p < 0.05$, clustered Wilcoxon signed-rank test). **e** Percentage of neurons with significant ($p < 0.05$) positive selectivity at each time-point. Shading indicates mean ± SEM across sessions. Dotted line indicates the expected percentage by chance

number of trials ($n = 1165$ cells across 8 sessions for PPC). Interestingly, we found that a large fraction of PPC neurons exhibited strong responses on Miss trials that were never apparent on CR trials (Supplementary Fig. 3; Miss vs. CR: $0.305 \pm 0.010$) or even on passively-presented Target trials. Together, these data argue that PPC responses cannot be parsimoniously explained as encoding purely motor-related signals.

**GLM reveals multiplexed sensory and motor signals in PPC.** We further used a generalized linear model (GLM)[20–22] to disambiguate the contributions of stimulus, task engagement, and motor action on single neuron calcium responses across both Passive and Engaged conditions. Responses of each V1 and PPC cell were modeled (see "Methods" section) as a linear combination of components that were time-locked to stimulus presentation or licking onset (Fig. 4). We quantified the performance of the GLM as the proportion of variance explained ($R^2$) for a separate test dataset not used for fitting (Fig. 4e). Using all

significantly task-responsive neurons, the average model performance was higher for V1 ($0.334 \pm 0.005$, $n = 1915$ cells) than for PPC ($0.190 \pm 0.003$, $n = 3524$; V1 vs. PPC, $p = 4.01 \times 10^{-5}$, clustered Wilcoxon rank-sum test).

We then assessed the relative contribution of stimulus, engagement, and motor model components for cells in each area, and illustrate the performance of the model on a few examples (Fig. 4a–d). Most V1 cells exhibited a strong stimulus component ($0.494 \pm 0.013$, component of $z$-scored calcium response, see "Methods" section), which reflected sensory drive on Passive trials, whether the neuron preferred the target (Fig. 4a) or the nontarget (Fig. 4b) stimulus. By contrast, PPC neurons had much weaker stimulus components (Fig. 4f, left; $0.117 \pm 0.013$; V1 vs. PPC, $p = 6.84 \times 10^{-5}$, clustered Wilcoxon rank-sum test). The engagement component of the GLM-reflected stimulus-specific signals that occurred exclusively on Engaged trials. This could be disambiguated from the stimulus-independent motor component, which would be present on both Hit and False Alarm (FA) trials. Individual PPC neurons varied in the relative contribution of

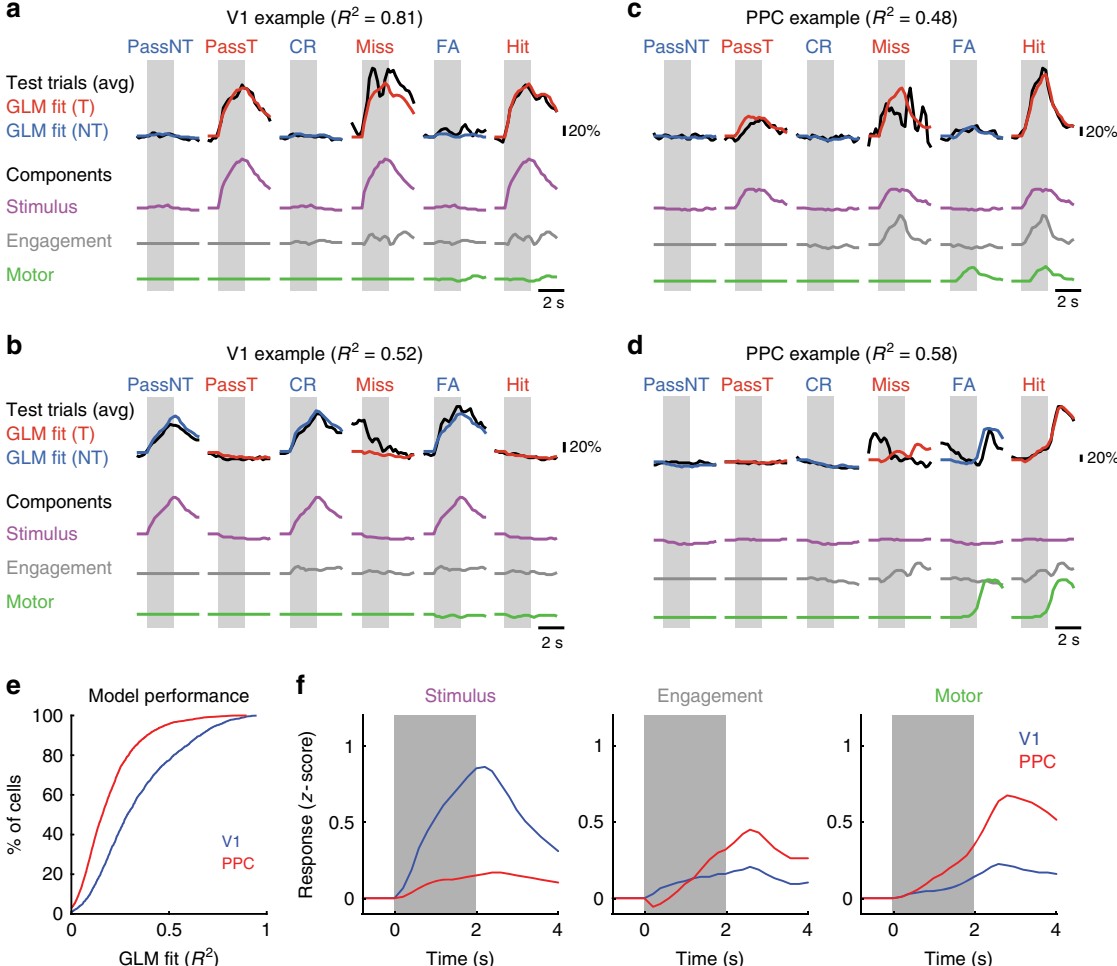

**Fig 4** Generalized linear model (GLM) of V1 and PPC responses. A generalized linear model was fit to training data to predict the calcium responses of individual V1 and PPC neurons using lagged predictors for stimulus, engagement and motor action (licking). **a** GLM model performance for an example V1 neuron. GLM fits on Target (red) and Non-target (blue) trials were compared with trial-averaged test data (black) across Passive (PassNT, PassT), Engaged No-Lick (Miss, FA), and Engaged Lick (FA, Hit) conditions. GLM components related to Stimulus (purple), Engagement (gray), and Motor (green) were summed to generate overall GLM fit. Shaded region represents time of stimulus presentation. **b** Same as **a**, but for a different, nontarget-preferring V1 neuron. **c**, **d** GLM model performance for two example PPC neurons. One neuron has a strong Engagement component (**c**), while the other has a stronger Motor component (**d**). **e** Cumulative histogram of model prediction performance on test trials, quantified for each cell as fraction of variance explained ($R^2$), for both V1 (blue, $n = 1915$ cells) and PPC (red, $n = 3524$). **f** Population-averaged GLM model components related to Stimulus (left), Engagement (middle), and Motor (right), for both V1 (blue) and PPC (red). For Stimulus and Engagement components, the preferred stimulus component for each cell was used for averaging. Calcium responses were $z$-scored before model fitting. Light shaded gray region demarcates duration of stimulus

engagement and motor components (Fig. 4c, d), but on average PPC cells exhibited a slightly (but not significantly) stronger engagement component (Fig. 4f, middle; V1, 0.183 ± 0.009; PPC, 0.233 ± 0.006; V1 vs. PPC, $p = 0.144$, clustered Wilcoxon rank-sum test) and a much stronger motor component (Fig. 4f, right; V1, 0.155 ± 0.005; PPC, 0.371 ± 0.005; V1 vs. PPC, $p = 0.015$, clustered Wilcoxon rank-sum test) as compared to V1 cells.

Linear models that consider the stimulus, task context, and licking together can explain some of the variance in calcium responses. But could even simpler models be sufficient? For each cell, we trained three additional partial models: a stimulus-only model, a motor-only model, and a stimulus + engagement model that was insensitive to the motor response (Supplementary Fig. 4). For V1 cells, the stimulus-only model performed nearly as well as the full model (median relative $R^2$: 0.905), but the motor-only model performed quite poorly (median relative $R^2$: 0.156). For the majority of V1 neurons, the stimulus + engagement model (which excluded licking terms) performed just as well as the full model (median relative $R^2$: 0.995; only 15.9% ± 5.3% of cells with worse fit). By contrast, while PPC neurons were better explained by the motor-only model (median relative $R^2$: 0.809) than by the stimulus-only model (median relative $R^2$: 0.353), both models

performed significantly worse than the full model for the majority of neurons (81.9% ± 6.2% of cells for stim-only, 55.9% ± 11.3% for motor-only). PPC neurons were reasonably well-explained by a stimulus + engagement model (median relative $R^2$: 0.910) but a large proportion of cells still exhibited worse fits compared to the full model (34.1 ± 10.7% of cells with worse fit). These results add further evidence that PPC encodes a combination of sensory and motor signals.

**PPC reflects both stimulus contrast and behavioral state.** Previous work has shown that in both primates[1,2] and rats[23], neurons in PPC encode not only the choice of the animal but also the amount of sensory evidence for that decision. To test whether neurons in mouse PPC similarly reflected the decision process, we varied the amount of sensory evidence from trial-to-trial by manipulating stimulus contrast. We also compared responses during engaged and passive conditions to examine how sensory and motor signals may be combined in PPC responses.

A subset of the mice ($n = 6$) were trained to perform a variant of the discrimination task, in which the contrast of the grating stimulus varied randomly from trial-to-trial (Fig. 5a). Mice

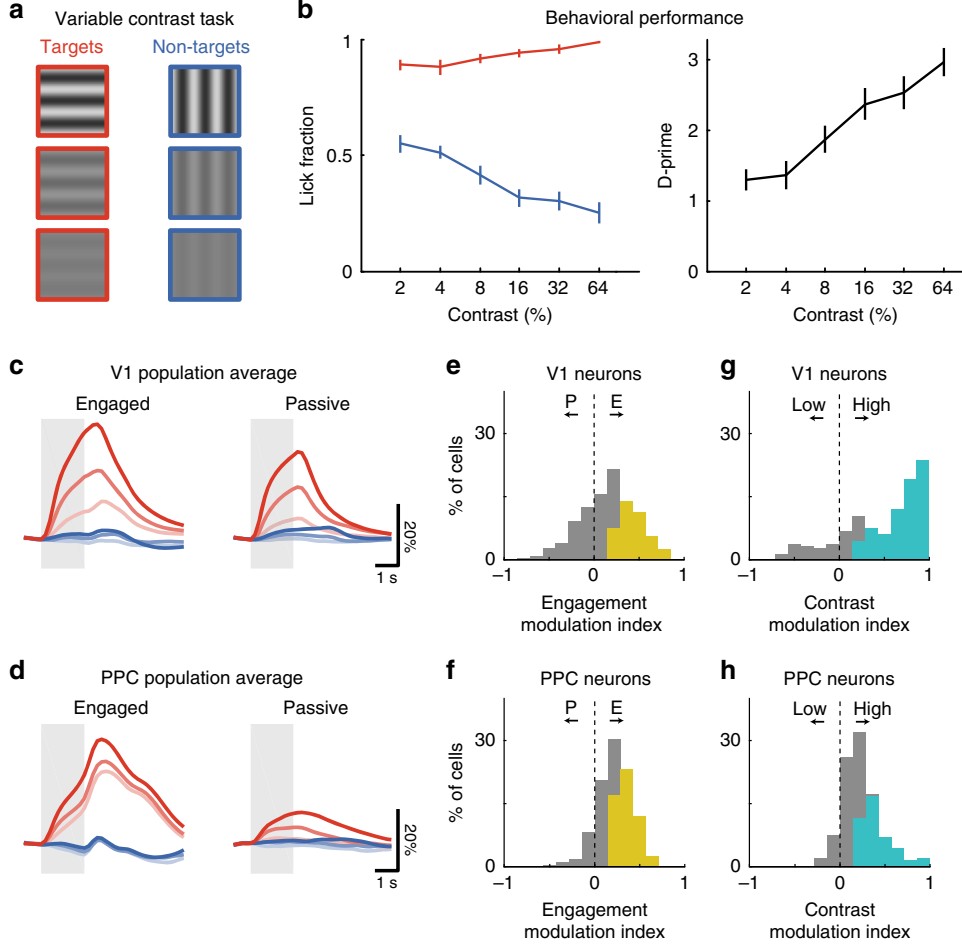

**Fig 5** PPC reflects both stimulus contrast and behavioral state. **a** Mice discriminated between orthogonally oriented Target and Non-target stimuli with contrast varying from 2 to 64%. **b** Behavioral performance (d-prime) on the variable contrast discrimination task, averaged across 19 sessions from 6 mice. Error bars indicate SEM. **c** Population trial-averaged response of all target-preferring V1 neurons ($n = 250$) across contrasts during correct Engaged (left) and Passive (right) target (red) and nontarget (blue) trials. Responses are averaged across low (2 or 4%, light shade), medium (8 or 16%, medium shade), and high (32 or 64%, dark shade) contrast. Light gray shaded regions demarcate duration of stimulus. **d** Same as **c** but for PPC ($n = 611$ target-preferring neurons). **e, f** Histograms comparing engagement modulation in V1 (**e**) and PPC (**f**), computed by comparing responses on Engaged vs. Passive high contrast trials. Colored bars indicate neurons with significant modulation. **g, h** Histograms comparing contrast modulation in V1 (**g**) and PPC (**h**), computed by comparing responses on high vs. low contrast Engaged trials. Colored bars indicate neurons with significant modulation

performed well above chance, even at very low contrasts ($d'$ at 2% contrast, $1.05 \pm 0.25$; $p = 0.031$, Wilcoxon signed-rank test), although performance degraded as contrast was lowered ($p = 1.10 \times 10^{-4}$, Friedman test) reflecting the decrease in the strength of sensory evidence (Fig. 5b). We imaged from neurons in V1 ($n = 8$ fields, 3 mice) and PPC ($n = 11$ fields, 6 mice) and computed contrast-response functions for both passive and engaged conditions. We again focused our analysis on target-preferring neurons (V1, $n = 250$; PPC, $n = 611$), which constituted the vast majority of task-responsive PPC neurons (95% in this dataset). Neurons were included for further analysis if they demonstrated significant Hit responses at multiple contrasts that could be well fit with a hyperbolic ratio function[24] (see "Methods" section).

As in the single-contrast task, the population response in V1 during the variable contrast task was robust in both Engaged and Passive conditions (Fig. 5c), with most neurons showing a significant response to at least one contrast during both conditions ($68.9\% \pm 11.8\%$ of cells). Conversely, PPC population activity was more robust in Engaged vs. Passive conditions (Fig. 5d), with only a subset of PPC neurons showing significant passive responses ($27.8\% \pm 14.9\%$; V1 vs. PPC, $p = 7.59 \times 10^{-3}$, Wilcoxon rank-sum test). The modulation of PPC activity by contrast cannot be explained by changes in motor behavior, as lick rate was unchanged as a function of contrast (Supplementary Fig. 5a).

We quantified the strength of modulation by contrast and engagement for each neuron using ROC-based indices that ranged from $-1$ to 1 (Fig. 5e–h). Engagement modulation index was similarly high in V1 and PPC (Fig. 5e, f; V1, $0.149 \pm 0.019$; PPC, $0.222 \pm 0.008$; V1 vs. PPC, $p = 0.209$, clustered Wilcoxon rank-sum test), but modulation by contrast was stronger on average in V1 compared to PPC (Fig. 5g, h; V1, $0.487 \pm 0.027$; PPC, $0.233 \pm 0.009$; V1 vs. PPC, $p = 0.021$, clustered Wilcoxon rank-sum test). Nonetheless, the mean contrast modulation index in PPC was significantly greater than zero ($p = 4.86 \times 10^{-3}$, clustered Wilcoxon signed-rank test).

We also analyzed the error trials to see whether stimulus and choice signals could be separately extracted from responses in PPC, and whether these signals depended on contrast (Supplementary Fig. 5). We compared False Alarm trials with Hit and Correct Reject trials, using imaging fields and sessions with at least five False Alarm trials at each contrast (7 of 11 fields, $n = 391$ neurons). We did not make comparisons with Miss trials given the low number of trials. The population response on FA trials was weak, but distinguishable from the response on CR trials (Supplementary Fig. 5a), especially during the response period (Supplementary Fig. 5b; FA greater than CR for all contrasts, $p < 0.05$, clustered Wilcoxon signed-rank test). Using an ROC-based approach, we again found that PPC encoded both stimulus and choice with differing time courses (Supplementary Fig. 5e; stimulus selectivity significant from 1.0 to 6.6 s for all contrasts, choice selectivity significant from 2.2 to 5.6 s for all contrasts, clustered Wilcoxon signed-rank test). Interestingly, some PPC cells encoded the stimulus in a contrast-dependent manner, but also encoded the choice in a contrast-independent manner (Supplementary Fig. 5c). We quantified the contrast-dependence of the selectivity index for each neuron by measuring its slope as a function of contrast (Supplementary Fig. 5d). A larger proportion of neurons exhibited significant contrast-dependence in stimulus selectivity (Supplementary Fig. 5f, $23.6\% \pm 19.9\%$ of cells) compared to the proportion with significant contrast-dependence in choice selectivity ($2.4\% \pm 1.6\%$). PPC may therefore simultaneously encode both contrast-dependent sensory signals and contrast-independent motor signals in the same population of neurons.

**Heterogeneous PPC responses to contrast and engagement.** PPC is on average modulated by both engagement and stimulus contrast, but closer examination of the individual PPC contrast-response functions revealed a great deal of heterogeneity. We therefore divided neurons into groups based on whether they exhibited significant modulation by contrast and/or engagement (Fig. 6a, b). A subset of PPC neurons ($21.2\% \pm 13.5\%$ of cells) showed strong modulation by contrast, but very weak modulation by engagement (Fig. 6c, d, left column). Such neurons therefore faithfully represented the sensory stimulus regardless of behavioral state. Conversely, a larger group of PPC neurons ($28.1\% \pm 12.3\%$) were gated by task engagement but showed little to no modulation with contrast (Fig. 6c, d, middle column). These neurons reflected the behavioral state and impending action of the animal irrespective of sensory drive. Lastly, a third subset of PPC neurons ($28.2\% \pm 10.6\%$) were significantly modulated by both contrast and engagement (Fig. 6c, d, right column). PPC therefore appears to contain both contrast-modulated "sensory" neurons as well as engagement-modulated "motor" neurons. This differs qualitatively from V1 (Fig. 6a, left; $p = 7.10 \times 10^{-17}$, $\chi^2$ test of independence, see also Supplementary Fig. 6), where most neurons ($74.8\% \pm 7.0\%$) are modulated by contrast, and much fewer by engagement alone ($9.2\% \pm 7.1\%$).

We then considered whether there was any anatomical organization of these functional properties within PPC. For each imaged PPC volume, functional subpopulations of contrast-modulated and engagement-modulated cells appeared to be intermingled across space (Fig. 6e). We computed the pairwise distance between all target-preferring neurons, and found no significant difference between within-group and across-group distances (Fig. 6f; within-group, $268 \pm 5$ μm; across-group $273 \pm 5$ μm; $p = 0.102$, clustered Wilcoxon signed-rank test). We also compared the functional properties of pairs of neurons as a function of distance, and found that the difference in contrast modulation or engagement modulation did not depend on distance (Fig. 6g; Pearson's correlation with distance, difference in contrast modulation index, $r = 0.003$, $p = 0.660$; difference in engagement modulation index, $r = 0.001$, $p = 0.960$). Therefore, PPC contains neurons with diverse visuomotor response properties that are spatially intermingled.

**PPC reflects choice independent of stimulus-reward contingency.** Most PPC neurons respond exclusively during target trials, while a smaller subset of neurons encode the sensory stimulus in a contrast-dependent manner. Error and GLM analyses suggest that these PPC populations encode choice and stimulus, respectively. However, because errors can reflect other factors such as impulsivity or inattention and thus be difficult to interpret, a more conclusive demonstration would require a clear dissociation of stimulus and choice. We therefore manipulated the stimulus-reward structure of the task and re-trained mice on a reversed sensorimotor contingency. By measuring activity from the same cells before and after reversal, we could test whether individual V1 and PPC neurons were more sensitive to stimulus identity or to the animal's choice.

After imaging the responses of neurons in V1 and PPC in the original go/no-go task, we reversed the reward contingencies of the stimuli (Fig. 7a). Licking in response to the original non-target stimulus (Stimulus B) was now rewarded with water, whereas licking to Stimulus A was punished with quinine. Three mice successfully learned the task after 7–11 days of training, although performance was slightly worse than before (Fig. 7b; d-prime, original, $2.96 \pm 0.50$; reversed, $1.51 \pm 0.22$). We then measured responses from the same populations of neurons in V1 and in PPC under the reversed reward contingency, and

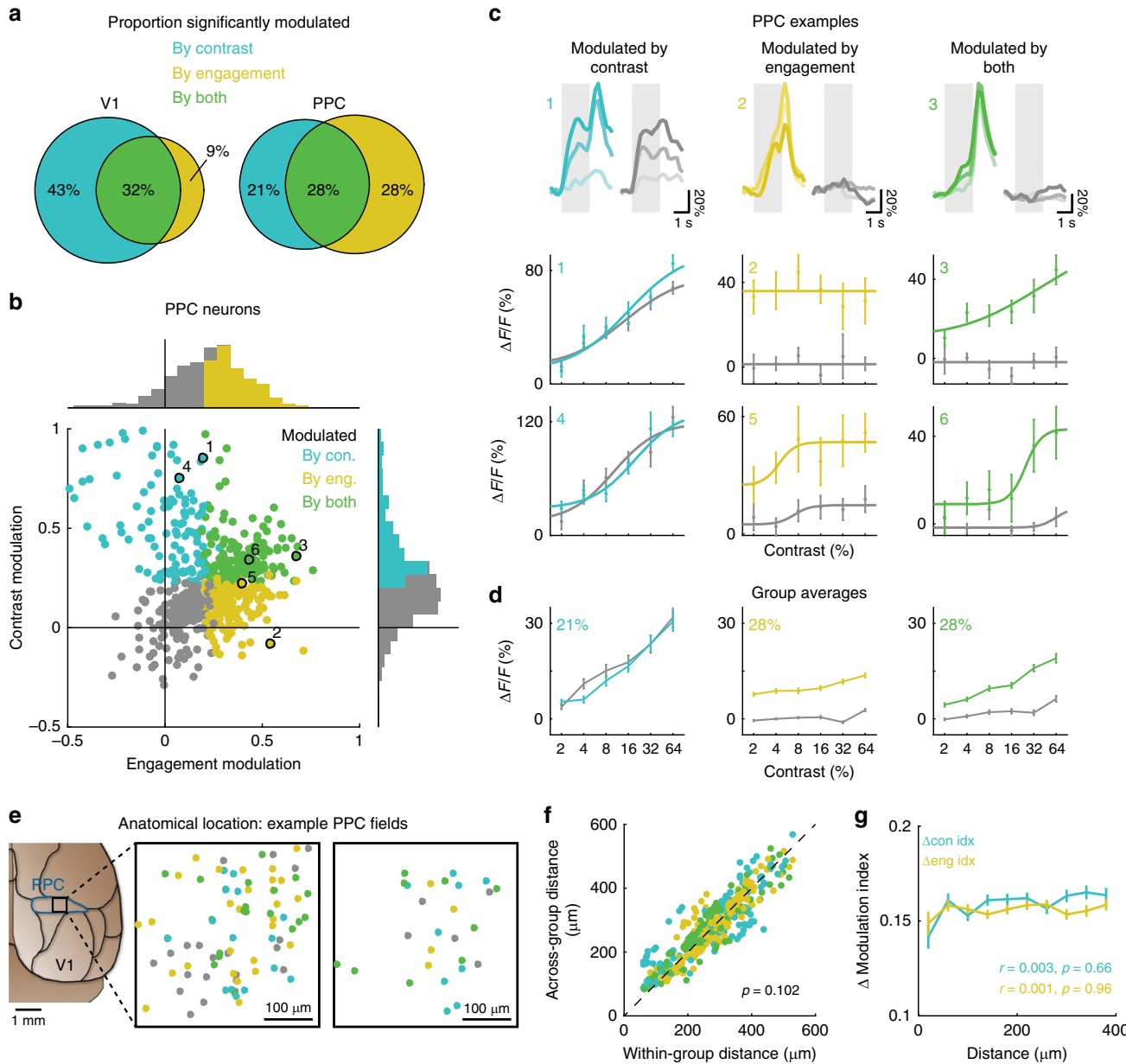

**Fig 6** PPC neurons are heterogeneous in their sensitivity to contrast and engagement. **a** Venn diagrams indicating proportions of target-preferring neurons in V1 and PPC with significantly positive modulation by contrast alone (cyan), engagement alone (yellow), or both contrast and engagement (green). **b** Scatter plot and histograms of contrast modulation vs. engagement modulation for all target-preferring PPC neurons ($n = 611$) imaged during the variable contrast task. Colored bars in histograms (same as Fig. 5f, h) indicate neurons with significantly positive modulation by contrast (cyan) or engagement (yellow). Colored dots on scatter plot demarcate neurons with significantly positive modulation by contrast alone (cyan), engagement alone (yellow), or both contrast and engagement (green). Individual examples in **c** are marked with the corresponding number. **c** Trial-averaged responses (top row) and contrast-response functions (middle and bottom rows) of example PPC neurons that were significantly modulated by contrast (left column), engagement (middle column), or both (right column). Contrast-response functions were computed from responses during the stimulus-period (gray shaded region in top row). Modulation index values for each example can be found by referring to **b**. Error bars indicate SEM across trials. **d** Group-averaged contrast-response functions. Percentages indicates proportion of PPC neurons within each group. Error bars indicate SEM across cells. **e** Left, approximate location of imaging fields within PPC (blue). Schematic modified from the Allen Mouse Brain Atlas (http://mouse.brain-map.org/static/brainexplorer). Right, relative spatial location of neurons with modulation by contrast, engagement, or both, in two example imaging sessions. Spatial locations of cells from 4 planes are collapsed into one. Scale bar, 100 μm. **f** Average within-group and across-group distances for each cell, colored by group. No significant difference was found for any individual group or for the whole population ($p = 0.102$, clustered Wilcoxon signed-rank test). **g** Difference in engagement (yellow) and contrast (cyan) modulation indices as a function of distance between cells. Error bars indicate SEM across cells. Distances beyond 400 μm are not shown. No significant correlation between modulation index and distance was found

identified the same neurons across imaging sessions using a semi-automated procedure[25] (see "Methods" section for details).

We analyzed the trial selectivity of individual neurons that had significant responses both before and after reversal (V1, $n = 488$

cells, 8 fields in 3 mice; PPC, $n = 509$ cells, 8 fields in 3 mice). Many neurons in V1 that were selective to a particular stimulus remained selective to the same stimulus after reversal, whether Stimulus A or Stimulus B (Fig. 7c). By contrast, many PPC

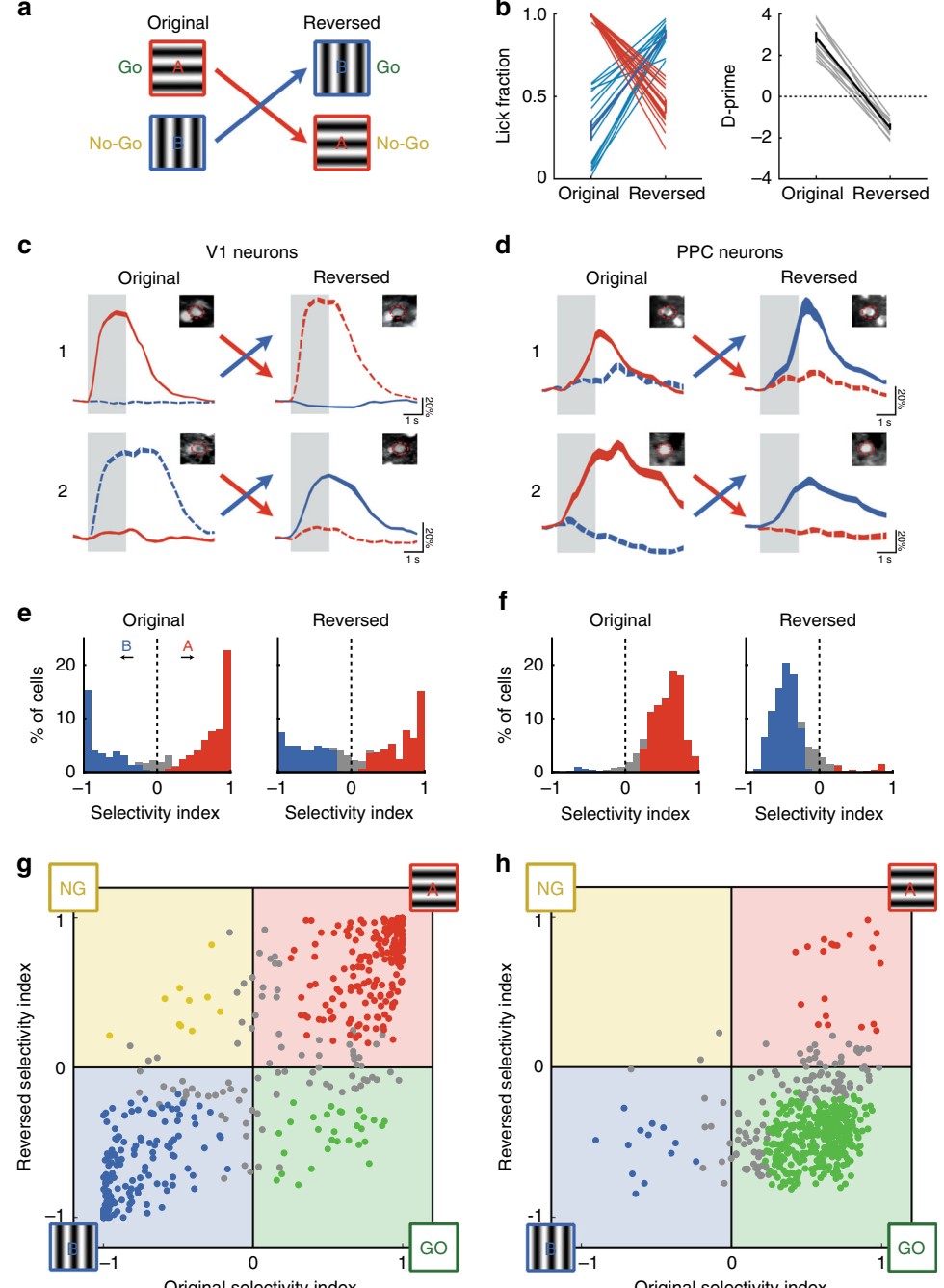

**Fig 7** PPC reflects choice independent of stimulus-reward contingency. **a** Mice were trained on a go/no-go discrimination task with a reversed reward contingency. **b** Behavioral performance before and after reversal. Left, response rate for Stimulus A (red) and Stimulus B (blue). Right, d-prime assuming Stimulus A as target. Performance after reversal was significantly different from zero ($p < 0.001$, Wilcoxon signed-rank test). **c** Neural response of two example V1 neurons before and after reversal of reward contingency. Colors indicate stimulus (red, A; blue, B) and line styles indicate whether the stimulus is target (solid) or non-target (dashed). Top row, response of a neuron selective to Stimulus A both before (left) and after (right) reversal. Bottom row, response of a neuron selective to Stimulus B. Insets indicate the average projection image before and after reversal, and ROI (dashed red line) that was used to extract the neural response. **d** Same as **c** but for two PPC neurons. These neurons show a switch in selectivity with reversal of reward contingency. **e** Histogram of selectivity index before and after reversal for V1 neurons with significant responses both before and after reversal ($n = 488$). Positive values indicate selectivity to the original Target, Stimulus A (red). Colored bars indicated neurons with significant selectivity. **f** Same as **e** but for PPC neurons ($n = 509$). **g** Scatter plot comparing selectivity index in V1 before and after reversal. Colored points indicate neurons with significant selectivity both before and after reversal. Neurons in the first and third quadrants are stimulus-selective, as they prefer Stimulus A (red) or Stimulus B (blue) both before and after reversal. Neurons in the second and third quadrants are choice-selective, as they prefer either the Go stimulus (green) or the No-Go stimulus (yellow) in both conditions. **h** Same as **g**, but for PPC

neurons did not exhibit stable stimulus selectivity, but instead appeared to track the animal's choice. These neurons preferred target stimulus A before reversal while preferring new target stimulus B after reversal (Fig. 7d). This strongly suggests that PPC neurons encode choice-related signals related to decision formation or motor planning.

Looking at the whole population, we measured the selectivity of neurons to Stimulus A trials before and after reversal. Stable selectivity would indicate sensory signals, whereas changes in this measurement would reflect signals related to choice. We found that V1 selectivity did not change significantly after reversal (Fig. 7e, before, $0.161 \pm 0.034$; after, $0.095 \pm 0.030$; $p = 0.256$, clustered Wilcoxon signed-rank test), with a relatively symmetrical distribution of neurons selective to either stimulus A or B. However, in PPC, this selectivity measure was dramatically altered with reversal of reward contingency (before, $0.513 \pm 0.013$; after, $-0.373 \pm 0.013$; $p = 4.15 \times 10^{-3}$, clustered Wilcoxon signed-rank test), with the majority of responsive neurons preferring the new target stimulus B after reversal.

We also compared the selectivity of individual neurons before and after reversal using a scatter plot (Fig. 7g, h). Purely stimulus-selective neurons will remain close to the unity line and in the first (bottom-left) and third (top-right) quadrants, whereas

choice-selective neurons will lie in either the second quadrant (top-left; for no-go selective cells) or the fourth quadrant (bottom-right; for go-selective cells). V1 neurons were strongly stimulus-selective, with the majority ($74.3\% \pm 5.2\%$ of cells, Fig. 8a, top) of neurons lying within the first and third quadrants (Fig. 7g). By contrast, the majority ($66.8\% \pm 13.0\%$ of cells, Fig. 8a, bottom) of PPC neurons were found in the fourth quadrant, indicating a preference for the rewarded target stimulus, regardless of its actual identity (Fig. 7h). These results indicate that the majority of PPC neurons reflected the choice of the animal.

**Task contingency reversal reveals distinct PPC subpopulations**. A small subset of PPC neurons ($6.9\% \pm 3.7\%$) did show stable stimulus selectivity before and after reversal. If PPC includes distinct populations of "stimulus" neurons and "choice" neurons, we hypothesized that they could be distinguished based on their responses on passive and error trials. Stimulus selectivity after reversal should be predictive of modulation strength with engagement and choice: with "stimulus" neurons showing robust passive responses and weak modulation by engagement and errors, and "choice" neurons expressing strong engagement and error modulation.

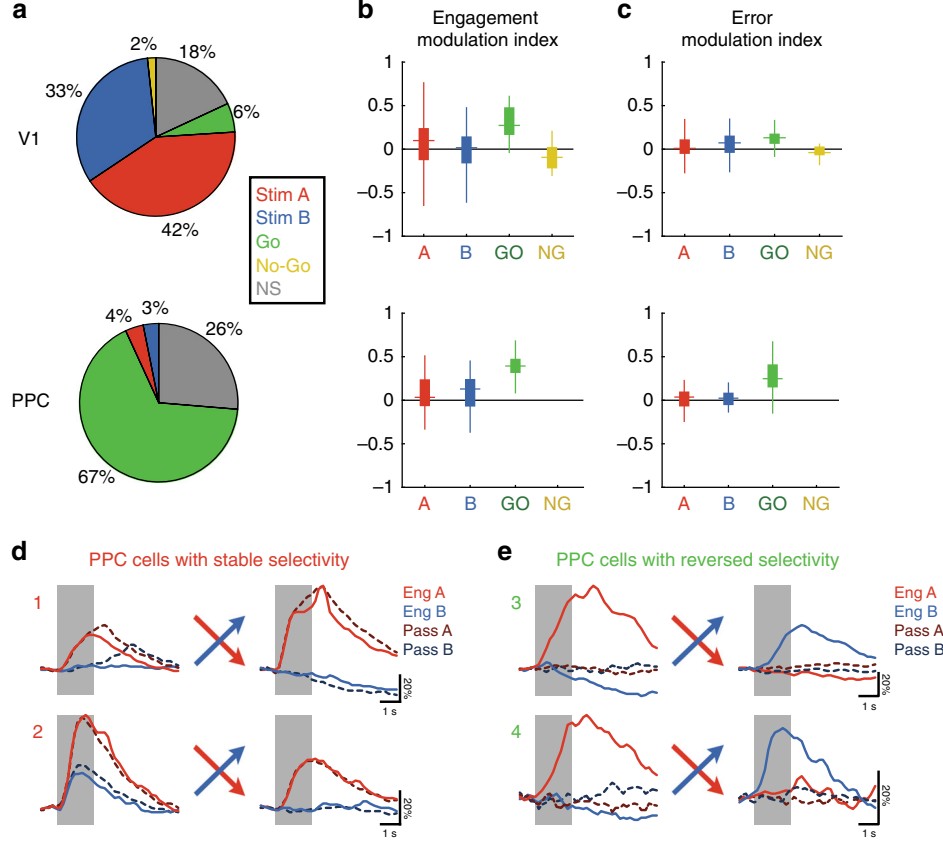

**Fig 8** PPC cells with reversed selectivity exhibit stronger modulation by engagement and by error trials. **a** Proportion of significantly responsive V1 (top) and PPC (bottom) neurons with stable stimulus selectivity (Stim A, 0°; or Stim B, 90°) or reversed selectivity (Go or No-Go). Cells with non-significant selectivity either before or after reversal are categorized as non-selective (NS). **b** Boxplots (with center line as median, box as interquartile range, and whiskers as 1.5 × IQR) of engagement modulation index for each response group, computed as the mean before and after reversal. "Go"-selective neurons (green) show stronger engagement modulation, i.e., weaker passive responses, than stimulus-selective neurons (red or blue). **c** Boxplots of error modulation index for each response group, computed by comparing False Alarm trials with Correct Reject trials both before and after reversal. "Go"-selective PPC neurons show stronger error modulation (green) than stimulus-selective neurons (red or blue). **d** Responses of two PPC neurons with stable selectivity to Stimulus A both before and after reversal of reward contingency. Colors indicate stimulus (red, A; blue, B) and dark, dashed lines indicate passive trials. These neurons show a robust passive response in both conditions, and have a low engagement modulation index. **e** Same as **d** but for two PPC neurons with reversed selectivity after reversal of reward contingency. These neurons show virtually no passive response in both conditions, and have a high engagement modulation index

To test this hypothesis, we measured the strength of engagement modulation in both stimulus-selective (to either Stimulus A or B) neurons and go-selective neurons, as defined by their selectivity before and after reversal (Fig. 8a). We found that go-selective PPC neurons had higher (near significant) modulation by engagement than stimulus-selective neurons (go-selective, $0.393 \pm 0.007$; stimulus-selective, $0.084 \pm 0.040$; $p = 0.065$, clustered Wilcoxon rank-sum test; Fig. 8b, bottom). Indeed, many PPC neurons that maintained selectivity to Stimulus A after reversal also had strong passive responses (Fig. 8d), and many neurons with reversed selectivity were strongly gated by engagement (Fig. 8e). We also tested to see whether a PPC neuron's responses on error trials related to its selectivity after reversal. Go-selective PPC neurons had significantly higher error modulation (computed by comparing False Alarm trials to Correct Reject trials) compared to stimulus-selective neurons (go-selective, $0.273 \pm 0.009$; stimulus-selective, $0.019 \pm 0.019$; $p = 0.040$, clustered Wilcoxon rank-sum test; Fig. 8c). Interestingly, the small subset of go-selective neurons in V1 ($5.9\% \pm 2.1\%$ of cells) also differed from stimulus-selective V1 neurons in their degree of engagement (Fig. 8b, top, $p = 0.016$) and error (Fig. 8c, top, $p = 0.021$) modulation, suggesting that these different properties are strongly related and can be used to distinguish functional cell types across different areas.

These findings demonstrate distinct subsets of neurons within PPC. One subset of neurons faithfully reflects the sensory input, both in passive conditions and after learning a new reward contingency. The larger proportion of PPC neurons, however, track the choice before and after re-training, and are strongly modulated by task engagement. The flexibility and heterogeneity of PPC responses suggests a role for PPC in the mapping of sensory inputs onto appropriate motor actions.

## Discussion

We developed a simple head-fixed visual decision task for mice with separate stimulus and response epochs, and used population imaging of single neuron responses to investigate the role of PPC in perceptual decisions. Our key findings are that PPC encodes both sensory and motor signals across a heterogeneous pool of neurons, and that its activity reflects task performance and demands. Together these results suggest that mouse PPC is responsible for neither pure sensory processing, nor for the control of motor output, but rather is important for the decision process itself – the process of mapping sensation to action.

The small size of the mouse brain has made it difficult to identify precise borders between different functional areas. PPC in the rodent, as classically defined by its thalamic inputs[26], is located in the region between anterior to V1 and posterior to somatosensory cortex. However, this location is essentially where both anatomical[11,12] and functional[13,14] mapping studies have identified the retinotopically-organized areas RL, A, and AM. The degree to which PPC overlaps with these secondary visual areas is a matter of debate. Some have argued that rodent PPC may have more in common with primate extrastriate cortex, given that inactivation specifically disrupts sensitivity on visual decisions[10]. Indeed, the stereotaxic coordinates used by us[3] and others[4,27] to target mouse PPC most directly overlap with area AM, which exhibits directionally-tuned responses even under anesthesia[14]. But is PPC simply a sensory visual area?

We have presented multiple pieces of evidence that point to a role for PPC beyond mere sensory processing. First, activity in PPC is strongly dependent on behavioral state, with only a minority of task-responsive PPC neurons exhibiting significant responses during passive viewing of stimuli (Fig. 2f). Secondly, the selectivity of PPC neurons is strongly biased toward target

stimuli (Fig. 2e). This bias is likely due to the asymmetry of the Go/No-go paradigm, and may reflect a learned association of stimulus and reward[28]. Third, PPC responses are modulated during error trials (Fig. 3, Supplementary Fig. 5). Information about the eventual choice of the animal can be decoded from the activity of PPC, as previously shown in both mice[3,4] and rats[5]. Finally, and most conclusively, the biased selectivity of most PPC neurons towards target stimuli is dramatically reversed when the animal is retrained on a different reward contingency (Fig. 7). Together, these results demonstrate that the stimulus-period responses in PPC are task-dependent and not purely sensory.

Alternatively, one may argue that the activity patterns we observed in PPC can be most parsimoniously explained as movement or motor-planning related signals. After all, movement-related activity would be present only during engagement, it would exhibit choice selectivity, and it would change with reward contingency. It is unlikely that PPC is directly involved in executing motor plans, as we have previously shown that optogenetic inactivation of PPC during the response period, or even during the delay between stimulus and response, has no effect on behavior[3]. However, the possibility remains that the PPC responses recorded in our task reflect planning- or movement-related signals that originate elsewhere. Although some PPC neurons (~30%) do show activity that appears motor-related, due to their contrast-independent nature (Fig. 6), we provide evidence that PPC is not a purely motor (or motor-planning) area either.

First, passive visual stimulation does induce a response in some PPC neurons (~20%), as previously shown in parietal area AM of anesthetized mice[14]. This subset of neurons tends to stably reflect the stimulus even with changes in reward contingency (Fig. 8). Second, we find a subset of PPC neurons that reflect the target stimulus on Miss trials when the animal fails to lick, even though such neurons are inactive during Correct Reject trials (Supplementary Fig. 3). Finally, the responses of many PPC neurons (~40%) shows modulation with stimulus contrast, even for the same decision and motor output (Fig. 6). This is reminiscent of previous primate[2] and rodent[23] PPC studies, where it has been shown that responses vary with the strength of incoming sensory evidence. These findings cohere with prior work in a delayed-response version of our task[3], in which choice-related selectivity was highest shortly after stimulus presentation and became weaker over the course of the trial. This argues against a motor-planning explanation which would instead predict increased choice coding with time.

In every task condition reported here, PPC responses were heterogeneous. A subset of PPC neurons have significant passive visual responses (~30%, Fig. 2f, E + P) and are modulated by contrast in both engaged and passive conditions (~20%, Fig. 6a, contrast). These neurons maintain their stimulus selectivity even after reversal of reward contingency (~10%, Fig. 8a, Stim). These "stimulus" neurons are spatially intermingled with the larger proportion of "choice" neurons that have task-gated responses (~70%, Fig. 2f, E only), weak contrast modulation (~30%, Fig. 6a, engagement), and choice-selective responses after contingency reversal (~70%, Fig. 8a, Go). Furthermore, a third group of neurons have multiplexed selectivity to stimulus, engagement, and motor signals, exhibiting complex responses to combinations of sensory or motor signals in engaged conditions. These cells exhibit modulation to both contrast and engagement (~30%, Fig. 6a, both) and their responses cannot be explained by partial GLM models that only use a subset of the task variables (~35%, Supplementary Fig. 4f).

Heterogeneous response properties have been previously reported in both primate[29,30] and rodent[5,31] PPC during decision tasks. Our work adds to this literature, and additionally provides

evidence that such heterogeneous responses exist in a spatially intermingled fashion within PPC of the mouse[32,33]. Do these response types form bona-fide cell classes or does PPC represent a category-free population, as others have proposed[5]? Although we do find that various properties (such as modulation by engagement, contrast, and reversal) are correlated with one another, more work needs to be done to determine whether functional neuronal subgroups are truly separable.

Recent studies in rodents have suggested that some of the heterogeneity observed in PPC responses may be due to encoding of information from recent experience, including information about past sensory stimuli[8], previous choices[9,27], and the presence or absence of reward. Such information about history could bias performance on sensorimotor tasks by providing animals with prior expectations about the value of particular stimuli[8] or actions[9]. While our findings do not allow us to directly address history-dependent biases in action selection, they are consistent with a role for PPC in selecting future motor actions.

Our results build on previous work in the field in several ways: First, we show that PPC neurons are actively gated by engagement in a sensorimotor task, with a substantial subpopulation of neurons exhibiting sensory responses only during task performance. Second, we used analysis of variable contrast stimuli and a GLM model to delineate the relative contributions of stimulus, motor preparation, and engagement to the responses of both V1 and PPC. These analyses showed that visual and motor signals are multiplexed within individual neurons. Finally, we showed that reversing reward contingencies causes the response selectivity to swap for the vast majority of PPC cells, indicating that the multiplexing of stimulus and response can be flexibly mapped depending on task contingencies. Taken together, these results bolster the evidence that PPC neurons are capable of mediating visuomotor transformations, although further evidence is necessary to establish this possibility conclusively.

The mouse posterior parietal cortex encodes behaviorally-relevant variables in a highly task-dependent manner, in analogy to prior work in primates. Our understanding of how decisions are computed and visuomotor transformations are made will be greatly aided by future circuit-level analyses of PPC function in this powerful model system[34].

## Methods

**Mice and surgery.** All experiments were carried out in mice of either sex using protocols approved by Massachusetts Institute of Technology's Animal Care and Use Committee and conformed to National Institutes of Health guidelines. Data were collected from adult (3–5 months old) wild-type (C57BL/6; $n = 15$) mice of either sex. The animals were housed on a 12-hour light/dark cycle in cages of up to 5 animals before the implants, and individually after the implants. All surgeries were conducted under isoflurane anesthesia (3.5% induction, 1.5–2.5% maintenance). Meloxicam (1 mg kg$^{-1}$, subcutaneous) was administered pre-operatively and every 24 h for 3 days to reduce inflammation. Once anesthetized, the scalp overlying the dorsal skull was sanitized and removed. The periosteum was removed with a scalpel and the skull was abraded with a drill burr to improve adhesion of dental acrylic. Stereotaxic coordinates for future viral injections were marked with a non-toxic ink and covered with a layer of silicon elastomer (Kwik-Sil, World Precision Instruments) to prevent acrylic bonding. The entire skull surface was then covered with dental acrylic (C&B-Metabond, Parkell) mixed with black ink to reduce light transmission. A custom-designed stainless-steel head plate (eMachineShop.com) was then affixed using dental acrylic. After head plate implantation, mice recovered for at least 5 days before beginning water restriction.

After behavioral training was complete, animals were taken off water restriction for 5 days before undergoing a second surgery to implant the imaging window. Procedures for anesthetic administration and post-operative care were identical to the first surgery. The dental acrylic and silicon elastomer covering the targeted region were removed using a drill burr. The skull surface was then cleaned and a craniotomy was performed over left V1/PPC, leaving the dura intact. Neurons were labeled with a genetically-encoded calcium indicator by microinjection (Stoelting) of 50 nl AAV2/1.Syn.GCaMP6s.WPRE.SV40 (University of Pennsylvania Vector Core; diluted to a titer of 10$^{12}$ genomes ml$^{-1}$) 300 μm below the pial surface. Between two and five injections were made in each exposed region, centered at V1 (4.2 mm posterior, 2.5 mm lateral to Bregma) and PPC (2 mm posterior, 1.7 mm

lateral to Bregma). Since the viral expression spreads laterally from the injection site, exact stereotaxic locations were photographed through the surgical microscope for determining imaging areas. Finally, a cranial window was implanted over the craniotomy and sealed first with silicon elastomer then with dental acrylic. The cranial windows were made of two rounded pieces of coverglass (Warner Instruments) bonded with optical glue (NOA 61, Norland). The bottom piece was a circular coverglass (4 mm diameter) that fit snugly in the craniotomy. The top piece was a larger circular coverglass (3–5 mm, depending on size of bottom piece) and was bonded to the skull using dental acrylic. Mice recovered for 5 days before commencing water restriction.

**Behavioral tasks.** We trained mice to perform a head-fixed go/no-go visual discrimination task, similar to previous designs[3]. Stimuli consisted of full-contrast sine wave gratings (spatial frequency: 0.05 cycles deg$^{-1}$; temporal frequency: 2 Hz) drifting at either 0° (upwards, target, Stimulus A) or 90° (rightwards, non-target, Stimulus B) away from vertical. Stimuli were presented to the right eye alone by placing the screen at an oblique angle to the animal. Behavioral training and testing was implemented with custom software written in Matlab (Mathworks) using Psychtoolbox-3[35] and Data Acquisition toolbox. Spout position was controlled by mounting the spout apparatus on a pneumatically-driven sliding linear actuator (Festo) controlled by two solenoids. Licks were detected using an infrared emitter/receiver pair (Digikey) mounted on either side of the retractable lick spout. Mice were water-restricted and earned most of their daily ration (1 mL) during training.

An auditory cue tone (5 kHz, 0.5 s, 65 dB SPL) indicated the beginning of each trial. After a 1 s delay, a visual stimulus was presented for 2 s. At the end of the stimulus epoch, the spout was rapidly moved within reach of the tongue, and remained within reach for 1.5 s. Correct licks during this period were rewarded with 5–8 μl water and a brief reward tone (10 kHz, 0.1 s). Licks to the non-target stimulus were punished with a white noise auditory stimulus alone (early training) or white noise plus 1–3 μl of 5 mM quinine hydrochloride in water (late training). This concentration was chosen to deter licking to non-targets without causing mice to lose motivation altogether (Supplementary Fig. 1). At the end of the response epoch, the spout was then rapidly retracted and remained out of reach until the next trial (3 s inter-trial interval).

Mice were trained in successive stages, as previously described[3]. Briefly, mice were first trained to lick a stationary lick spout during presentation of the target stimulus, and then non-target stimuli were gradually introduced. Spout withdrawal was introduced once mice showed good discrimination performance ($d' > 1$ and $R_{HIT} - R_{FA} > 30\%$ for consecutive sessions), with the spout initially being extended before stimulus onset, but gradually delayed to extend after stimulus offset. Once mice reached high levels of performance at the final stage of the task ($d' > 1.5$ and $R_{HIT} - R_{FA} > 50\%$), they were removed from water restriction for window implantation. Mice reached criterion performance after an average of 92 ± 11 sessions. After recovery from window implantation surgery, they were re-trained to a level of high performance (2–7 days) before beginning experimental sessions. Any sessions with poor performance were discarded (minimum performance criterion: $d' > 1$ and $R_{HIT} - R_{FA} > 30\%$).

During imaging experiments, blocks of engaged behavior trials were alternated with blocks of passive viewing. Blocks were 5–10 min in duration (40–80 trials per block). During passive blocks, the spout was out of reach for the duration of the block. A few extra passive trials were given (without imaging) before each passive block to ensure that mice did not expect spout presentation during all imaged passive trials. The sequence of target and non-target stimuli presented for a given passive block was matched to the sequence of stimuli used for the preceding engaged block. In some cases, instead of alternating between engaged and passive blocks, the engaged blocks were all grouped together at the beginning of the session, followed by an equal number of consecutive passive blocks. No difference in results was observed for alternating vs. grouped blocks.

Some mice ($n = 6$) were trained on a variable contrast version of the task. On each trial, the stimulus was randomly set to one of six contrasts (2, 4, 8, 16, 32, or 64%), regardless of whether the stimulus was a target or non-target. The mouse therefore could not predict the contrast of the stimulus from trial to trial.

Some mice ($n = 3$) were re-trained after initial imaging experiments on a reversed reward contingency. Reward contingency was switched abruptly, with reward given for licks to Stimulus B and no reward for licks to Stimulus A. Because mice were quickly discouraged by the reversal, no quinine punishment was initially given. Additionally, the reward tone (10 kHz, 0.1 s) was paired with the onset of the new target stimulus (Stimulus B) early in re-training, in order to encourage licking. Three of the five mice trained on this reversed contingency achieved criterion performance after re-training for 10 ± 2 days; the other two were removed from the study.

For all tasks, behavioral d-prime ($d'$) was computed by norminv(Hit rate) – norminv(False alarm rate), where norminv() is the inverse of the cumulative normal function[34,36]. Values of Hit and False alarm rate were truncated between 0.01 and 0.99, setting the maximum $d'$ to 4.65. For illustration purposes, d-prime before and after reversal was computed using Stimulus A as the target in both conditions (Fig. 7b).

**Two-photon imaging.** GCaMP6s fluorescence was imaged 14–35 days after virus injection using Prairie Ultima IV 2-photon microscopy system with a resonant

galvo scanning module (Bruker). For fluorescence excitation, we used a Ti-Sapphire laser (Mai-Tai eHP, Newport) with dispersion compensation (Deep See, Newport) tuned to $\lambda = 910$ nm. For collection, we used GaAsP photomultiplier tubes (Hamamatsu). To achieve a wide field of view, we used a $16\times$ / 0.8 NA microscope objective (Nikon), which was mounted on a z-piezo (Bruker) for volume scanning. An optical zoom of $2\times$ was used in most cases to improve spatial resolution. Resonant scanning (15.9 kHz line rate, bidirectional) was synchronized to z-piezo steps in the acquisition software for volume scanning. For volume scanning, four $441 \times 512$ pixel imaging planes separated by 20 or 25 μm were imaged sequentially at a stack rate of 5 Hz in 5–10 min imaging blocks. There was very little redundant sampling of neurons between imaging planes (<1%) as assayed by correlation coefficient of spontaneous activity. Laser power ranged from 40–75 mW at the sample depending on GCaMP6s expression levels. Photobleaching was minimal (<1% min$^{-1}$) for all laser powers used. A custom stainless steel plate (eMachineShop.com) attached to a black curtain was mounted to the head plate before imaging to prevent light from the visual stimulus monitor from reaching the photomultiplier tubes. During imaging experiments, the polypropylene tube supporting the mouse was suspended from the behavior platform with high tension springs (Small Parts) to dampen movement.

**Image preprocessing and cell selection.** Calcium imaging data were acquired using PrairieView acquisition software and sorted into multi-page TIF files. All analyses were performed using custom scripts written either in ImageJ or MATLAB (Mathworks).

Images were first corrected for $X$–$Y$ movement by registration to a reference image (the pixel-wise mean of all frames) using 2-dimensional cross correlation. To identify responsive neural somata, a pixel-wise activity map was calculated as previously described[37]. Neuron cell bodies were identified using local adaptive threshold and iterative segmentation. Automatically-defined ROIs were then manually checked for proper segmentation in a MATLAB-based graphical user interface (allowing comparison to raw fluorescence and activity map images). To subtract the influence of local neuropil on somatic signals, the fluorescence in the somata was estimated as $F_{corrected\_soma}(t) = F_{raw\_soma}(t) - 0.7 \times F_{neuropil}(t)$, where $F_{neuropil}$ was the defined as the fluorescence in the region 0–15 mm from the ROI border (excluding other ROIs)[17]. $\Delta F/F$ for each neuron was calculated as $\Delta F / F_t = (F_t - F_0) / F_0$, with $F_0$ defined as the mode of the raw fluorescence density distribution.

To align ROIs between different imaging sessions across days (Figs. 7 and 8), we used a semi-automated method similar to prior work[25]. First, for each plane, anchor points were manually defined by visual comparison of the two average projection images. These anchor points helped to define a predicted displacement vector field that would be used to map coordinates from one session to the other. For each coordinate, the predicted vector was defined by the average (weighted inversely by distance) of the vectors for all defined anchor points.

Next, for each ROI, a square region (~$4 \times$ the size of the ROI) around the ROI was selected. To determine the displacement across sessions, we computed the normalized cross-correlation of this square with the average projection of the other session. This was multiplied point-by-point with a mask that decayed gradually with distance from the predicted displacement vector, and then smoothed with a 2-D Gaussian filter. The peak of the resulting image was taken to be the actual displacement vector of the ROI. This process biases the displacement of each ROI towards the vector predicted from the manually defined anchor points. Finally, any ROIs with a computed displacement vector that differed by greater than 5 pixels from the predicted vector were flagged for manual inspection, and then either redrawn or removed.

After image preprocessing and $\Delta F/F$ extraction, traces were sorted by trial type (Hit, Miss, Correct Reject, False Alarm) and condition (Engaged, Passive). The baseline response (1 s before stimulus onset) was subtracted from each trial. A neuron was considered task responsive if its mean $\Delta F/F$ during the last 1.6 s (8 frames) of the stimulus period was significantly ($p < 0.01$, $t$-test) greater than the pre-stimulus baseline (1 s), for either hit or correct reject trials. Neurons also had to meet a signal-to-noise criterion, needing a trial-averaged response that exceeded a threshold of at least two standard deviations above baseline during either the stimulus or response period. All further analyses are based on responses during the last 1.6 s of the stimulus period, unless noted otherwise. Cell selection criteria for error analyses, variable contrast analyses, and reversal analyses are described in the appropriate sections below.

**Statistics.** The data were obtained from 15 mice, 5 with V1 only, 6 with PPC only, and 4 with both V1 and PPC. For most mice, multiple fields of view were sampled within V1 or within PPC. For mice with both V1 and PPC windows, fields from each area were sampled on interleaved sessions. Each field was imaged for a single session, consisting of multiple Engaged and Passive blocks and yielding on average 94 trials (minimum 47) per stimulus per condition. For variable contrast tasks, an average of 25 trials (minimum 8) were acquired per contrast. Significantly task-responsive cells from different fields were pooled by area. No tests were conducted to determine sample size. For the full-contrast task, the data came from 1915 V1 cells (18 fields, 9 mice) and 3524 PPC cells (22 fields, 10 mice). For the variable-contrast task, the data came from 250 V1 cells (8 fields, 3 mice) and 611 PPC cells

(11 fields, 6 mice). For comparisons before and after contingency reversal, the data came from 488 V1 cells (8 fields, 3 mice) and 509 PPC cells (8 fields, 3 mice).

All statistical analysis was performed using custom-written scripts in MATLAB or R. In all cases, data was not assumed to be normal, and nonparametric and/or permutation tests (2000 permutations) were used to assess statistical significance of results. All tests were two-tailed, and a significance level of $p < 0.05$ was considered significant. Unless otherwise noted, all measures are reported as mean ± SEM. When estimating the percentage of selective or modulated neurons, bootstrapping across imaging fields was used to generate confidence intervals on the percentages for each area. When testing the statistical significance of differences between V1 and PPC neurons that were pooled across imaging fields, clustered nonparametric tests[38,39] were used to account for intra-cluster correlations[31,40] using the *clusrank* package in R.

**Selectivity and error trial analyses.** Neurons were marked as target- or non-target-preferring (Fig. 2e) based on their mean response during Engaged trials. Neurons were marked as task-gated if they did not exhibit a significant response to their preferred stimulus during Passive trials (Fig. 2f).

All comparative indices (engagement modulation index, error modulation index, contrast modulation index, selectivity index) were computed using a receiver operating characteristic (ROC) analysis, which quantifies the ability of an ideal observer to discriminate between trial types based on single trial responses[18,36]. Each index was derived from the area under the ROC curve (auROC), and defined as $2\times(auROC-0.5)$; this value ranged from $-1$ to $1$[5]. Unless otherwise noted, comparative indices were computed by comparing the stimulus period response (averaged over the last 1.6 s). To determine whether comparative indices were significant for individual neurons, we used a permutation test. We shuffled the labels for each trial and recomputed the index 2000 times to create a distribution of indices that could have arisen by chance. Indices outside the center 95% interval of this distribution were considered significant ($p < 0.05$).

For error trial analyses, only behavioral sessions and imaging fields with at least five trials per condition were used. Consecutive miss trials occurring at the end of a session were excluded, as these trials are confounded by lack of motivation. Analysis was additionally limited to Hit-preferring neurons. For analysis of False Alarm trials (Fig. 3), we analyzed 1053 V1 cells from 16 fields, and 3034 PPC cells from 21 fields. For analysis of Miss trials (Supplementary Fig. 3), we analyzed 202 V1 cells from 3 sessions, and 1165 PPC cells from 8 fields.

We assessed "stimulus-related" selectivity by comparing Hit to FA trials or Miss to CR trials. We define "choice-related" selectivity as any premotor- or decision-related signal, assessed by comparing FA to CR trials or Hit to Miss trials (Supplementary Table 1). For most comparisons, any selectivity measured after the stimulus period may include signals related to reward, punishment, or motor output. We therefore restricted analyses of selectivity to the stimulus period.

Selectivity was computed across time using a ROC-based index evaluated independently at each time bin (200 ms)[19,29]. A clustered Wilcoxon signed-rank test[39] was used at each time bin to test whether average selectivity for each area was significantly different from zero ($p < 0.05$, no correction for multiple comparisons).

To assess the dependence of PPC responses on motor output (Supplementary Fig. 2), we noticed that a few animals exhibited natural variability in licking behavior across False Alarm (FA) trials. We analyzed neurons from the sessions (846 PPC cells from 3 fields) that had at least five FA trials with 1–3 licks and five FA trials with 5 or more licks. We additionally randomly selected Hit trials with the same number of licks (5–9) to match the average number of licks in the FA (5–9) condition. Selectivity between these conditions was evaluated using a ROC-based index as above.

**Generalized linear models.** We used generalized linear models (GLM) to regress recorded calcium signals against a time series of task events[20–22]. Calcium responses for each cell were z-scored and modeled as the linear combination of various task events, each convolved with a filter:

$$
\begin{aligned}
\hat{y}_t = {} & \beta_0 + \sum_{i=0}^{20} \beta_i^{S_{tar}} x_{t-i}^{S_{tar}} + \sum_{i=0}^{20} \beta_i^{S_{nt}} x_{t-i}^{S_{nt}} + \beta_0^E x^E \\
& + \sum_{i=0}^{20} \beta_i^{E_{tar}} x_{t-i}^{E_{tar}} + \sum_{i=0}^{20} \beta_i^{E_{nt}} x_{t-i}^{E_{nt}} + \sum_{i=-10}^{15} \beta_i^{L} x_{t-i}^{L}
\end{aligned}
\tag{1}
$$

The response of a neuron at frame $t$ is modelled ($\hat{y}_t$) by the sum of a bias term ($\beta_0$) and the weighted ($\beta_i$) sum of various additional binary predictors at different lags ($i$). Binary predictors for the target stimulus ($x_t^{S_{tar}}$) and non-target stimulus ($x_t^{S_{nt}}$) indicated the onset of stimulus presentation in either engaged or passive trials. Binary predictors for engagement included a constant offset ($x^E$) that was 1 during engaged trials and 0 otherwise, as well as stimulus predictors ($x_t^{E_{tar}}$, $x_t^{E_{nt}}$) that indicated the duration of stimulus presentation during engaged trials only. Binary predictors ($x_t^L$) for licking indicated the duration of lick bouts, which were defined as groups of licks with an inter-lick interval <1 s. The number of lags were chosen to capture both the duration of the event (2 s for stimulus, 1 s for licking) as well as the slow offset dynamics of the calcium response[17]. Lags were chosen to be strictly positive (causal) for stimulus and engagement predictors, but both positive and negative (anti-causal) for licking predictors. The final model had 112 coefficients including a constant bias term. Models were fit for all task-responsive neurons,

where the first 5 s from each trial (following auditory cue onset) was extracted and concatenated for analysis.

Models were fit using ridge regression using procedures similar to that of previous studies[20,22,41]. We first set aside 20% of trials from each condition (Hit, Correct Reject, Miss, False Alarm, Passive Target, Passive Nontarget) for testing. A regularization parameter $\lambda$ was estimated for each cell from among a range of $\lambda$ values ($10^{-2}$ to $10^{4}$) using cross-validation. Model performance was measured by computing the proportion of explained variance, or the coefficient of determination ($R^2$). Five-fold cross-validation was used to choose the $\lambda$ that maximized the performance on the remaining 80% of training trials. The final model for each cell was fit using the best $\lambda$, and then performance was evaluated by measuring $R^2$ of predictions for the holdout test set.

The strength of each model component (stimulus, engagement, motor/licking) was evaluated for each V1 and PPC cell (Fig. 4f) by setting all other model coefficients to zero and finding the model predictions using just one component, averaging across both training and test trials. Model components were therefore in units of z-scored calcium response and can be directly compared with calcium response waveforms. For stimulus and engagement components, the preferred trial type (Hit or Correct Reject) was used for each cell, whereas for the motor component, Hit trials were used for all cells. To evaluate the relative strength of the various components across V1 and PPC cells, the component strength for each cell was averaged over a window of 0 to 4 s after stimulus onset, and then a Wilcoxon rank-sum test was applied.

We also trained GLM models with a subset of the task predictors (Partial GLM models, Supplementary Fig. 4) using the same procedures. Three additional models were trained for each cell: a stimulus-only model which included the stimulus and constant bias terms (43 coefficients), a motor-only model which included only the licking and bias terms (27 coefficients), and a stimulus + engagement model which used all terms except licking (86 coefficients). A separate cross-validation process and regularization parameter was computed for each model, but the same training and test trials were using to evaluate performance.

Performance on the partial models was evaluated only for cells with significant fits ($p < 0.05$) to the full model. The test trial dataset was bootstrapped 2000 times, and a $p$ value was computed for each cell as the fraction of iterations with $R^2 \leq 0$. ($R^2$ can be negative if the model performs worse than a horizontal line on the held-out test data). The same bootstrapped datasets were also used to determine whether each partial model performed significantly above chance ($p < 0.05$), or significantly worse than the full model fit ($p < 0.05$).

**Contrast task analysis**. For data acquired during the variable contrast task, neurons were considered significantly responsive if the mean $\Delta F/F$ during the stimulus period was significantly above threshold for at least two of the six contrasts of the same stimulus. We focused our analyses on target-preferring neurons, which were included if their mean response across contrasts was greater for Hit (target) trials compared to Correct Reject trials.

Single neuron contrast-response functions were fit to the hyperbolic ratio function, also known as the Naka-Rushton function[24]:

$$R(C) = R_{max} \frac{C^n}{C^n + C_{50}^n} + R_0 \qquad (2)$$

where $R(C)$ is the neural response as a function of contrast, $R_{max}$ is the saturation point, $C_{50}$ is the contrast at the half-saturation point, $R_0$ is the baseline response, and $n$ is an exponent that determines the steepness of the curve. The responses for both Engaged and Passive conditions were fit simultaneously, with $n$ constrained to be constant across conditions, by minimizing the sum (across data points) of the squared error between the model and the data, divided by the variance of that data point.

We evaluated the goodness of fit for each neuron using a bootstrap hypothesis test, as detailed by others[42]. Briefly, we tested the null hypothesis that the mean of the probability distribution underlying the neural responses was identical to the predictions of the model. We measured the observed prediction error ($e_{obs}$) and computed the probability of observing an error at least as large if the null hypothesis were true. To sample from a distribution that conformed to the null hypothesis, we shifted the data such that the mean responses equaled the model predictions, and drew 1000 bootstrap samples from this dataset, computing a prediction error ($e_i$) for each. The proportion of samples for which the prediction error was larger than $e_{obs}$ is the achieved significance level. For neurons with an achieved significance level below 10% ($p < 0.1$), there was sufficiently strong evidence against the model, and therefore these neurons were excluded from further analysis.

Contrast modulation index was computed to compare Engaged trial responses on high (64%, 32%) vs. low (2%, 4%) contrasts, using a renormalized ROC index which ranged from $-1$ to 1, as described above. Engagement modulation index was computed by comparing Engaged vs. Passive trials, using high contrast trials only. A permutation test was used to assess significance ($p < 0.05$) of these indices by shuffling trial labels 2000 times and comparing the measured index to the shuffled distribution of indices.

To assess anatomical spatial organization, pairwise Euclidean distance was measured between all task-responsive, target-preferring neurons in the same imaged volume (Fig. 6e–g). Cells were grouped into four groups based on whether

they were contrast-modulated, engagement-modulated only, both, or neither, and then the average within-group and across-group distance was computed for each cell (Fig. 6f). To compute differences in modulation index as function of distance (Fig. 6g), pairs of cells were binned by distance in 40 μm bins.

Selectivity on error trials (Supplementary Fig. 5c-f) was computed by comparing FA trials to Hit and CR trials (with matched contrast) using a frame-by-frame ROC analysis, as described above. The contrast-dependence of the auROC-based index was estimated by regressing the index against log contrast, and then finding the slope. The significance ($p < 0.05$) of the slope was estimated for each cell by shuffling the trial labels 2000 times for each contrast and then computing the auROC on the shuffled data to generate a distribution of 2000 slopes.

**Reverse contingency task analysis**. We imaged from the same field of neurons before and after reversal of reward contingency. The two sessions were separated by an average time interval of $16 \pm 1$ days. A semi-automated method was used to align ROIs between the two sessions (see "Image preprocessing and cell selection" section). Neurons were included for analysis only if a significant response ($p < 0.01$) to either stimulus was observed both before and after reversal. Selectivity index was computed separately before and after reversal training (Fig. 7e–h) by comparing responses to the Stimulus A (original target) and Stimulus B (original non-target), with positive values indicating preference for the Stimulus A. Separate permutation tests (2000 iterations) were used to assess significance of the selectivity before and after reversal. Neurons with significant selectivity ($p < 0.05$) both before and after reversal were categorized (Fig. 8a) based on the sign of the selectivity, and whether it was stable or altered with reversal. Engagement modulation index (Fig. 8b) for each neuron was computed separately before and after reversal, comparing Engaged and Passive responses using the neuron's preferred stimulus, and then taking the mean of the two values. Error modulation index (Fig. 8c) for each neuron was also computed separately before and after reversal, comparing FA and CR responses, and then taking the mean of the two values.

**Data availability**. The data and custom MATLAB analysis code that support the findings of this study are available from the corresponding authors upon request.

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

## Acknowledgments

This work was supported by the NIH (M.J.G., F32-EY023523 and K99-MH104259; M.S., R01-EY007023 and U01-NS090473); the NSF (G.N.P., Graduate Research Fellowship; M. S., EF1451125); the Simons Center for the Social Brain (M.S.), and the Picower Institute Innovation Fund (M.J.G.; M.S.). We thank J. Sharma, A. Boesch, V. Li, C. Le, J. Krizan, V. Breton-Provencher, and T. Emery for technical assistance; R. Huda, M. Hu, H. Sugihara, and C. D. Harvey for helpful discussions and/or comments on the manuscript; L. L. Looger, J. Akerboom, D. S. Kim, and the Genetically-Encoded Neuronal Indicator and Effector (GENIE) Project at Janelia Farm Research Campus Howard Hughes Medical Institute for generating and characterizing GCaMP6 variants.

## Author contributions

G.N.P. and M.J.G. designed experiments with input from M.S. M.J.G. performed the surgeries. M.J.G. and G.N.P. performed the imaging experiments. M.J.G., G.N.P., J.W., and B.C. performed the behavioral training. G.N.P. developed the behavior software. G.N.P. analyzed the data with comments from M.J.G. and M.S.. G.N.P., M.J.G., and M.S. wrote the manuscript.

## Additional information

**Competing interests:** The authors declare no competing interests.

