## [Peer Review File · Nature Communications]

Reviewers' comments:

Reviewer #1 (Remarks to the Author):

Pho et al studied the function of PPC in a visual go-no go task and found that PPC appears to encode a mix of sensory and motor features. The authors used a combination of approaches including the analysis of error trials, modulation of the sensory stimulus, and reversals to aid understanding what is encoded in PPC. The biggest weakness of the paper is the task design, in which many behavioral variables are intermixed, like task engagement/attention and motor response (that is, the mouse only has to pay attention and create a motor response following the 'go'-related cue). I do not see any way for the authors to fully overcome this shortcoming. The use of error trials and miss trials are helpful but often it is hard to interpret error trials because it is unclear why the animal got the trial incorrect. I worry a bit that some behavioral differences between engaged/not engaged or contrast conditions could underlie the encoding the authors see in PPC. The authors have tried to control for things like licking, but I really do not see any way to eliminate or fully control for potential behavioral confounds given the go-no go task design. I do not have any major suggestions for improvement because the authors have done the best they can with the data they have and it would be unreasonable to ask for data from a better-designed behavioral task. Therefore, I am confident that the authors have done a reasonable job to try to overcome the limitations that arise from their task design. The results of PPC as a sensory and motor area likely will not greatly change how the field thinks about PPC. For me, those results confirm the majority view in the field. However, there is a minority of people who think of PPC as an area for visual representations, so perhaps the paper will be beneficial for that audience. From the authors' introduction, the main debate in the field appears to be whether PPC is a sensory-to-motor area for senses other than vision, which is not addressed here. Despite the limitations of the interpretation that is possible from the task design and despite the lack of an entirely novel result, the findings, in my opinion, will be useful for the field.

Reviewer #2 (Remarks to the Author):

Pho et al imaged neurons in mouse PPC and V1 during a go/no-go visual discrimination task to test the hypothesis that mouse PPC is critical for visual-motor transformations. The authors suggest that this modulation gives rise to flexible and heterogeneous neuronal activity within the PPC that is distinct from activity modulation in V1, which seems to be primarily driven by visual input. The use of a paradigm where visual stimuli and the motor response are close together in time, and the temporal resolution of imaging makes it very difficult to disentangle planning and motor signals; however, the authors used a variety of manipulations to provide evidence for a PPC signal that does not appear to be purely sensory (particularly in contrast to V1 responses during the same task), nor purely motor. While it would be ideal to provide further evidence that the observed activity is not motor related (e.g., temporal profile centered on the initial motor response showing activity following sensory related activity but preceding the actual motor response), such analyses are beyond the scope of the present paper that mostly acknowledges the boundaries of the evidence presented here.

Major:

1) Some visual representation of the anatomical location of the data presented here would be very helpful, particularly given that the first point in the discussion is related to the anatomy of this region of the mouse parietal cortex. For example, could the data shown in Figure 6e be shown on a top down view of the mouse dorsal cortex?

2) Why isn't the population number (e.g., median r-squared) reported for the stimulus+engagement

model for PPC (but is for V1)? Wouldn't this be a critical factor for the argument that PPC is not motor or sensory?

3) Evidence for transformation of sensory information, other than visual information, into motor commands is not presented in this paper. Particularly given the current state of the literature suggesting differences with other sensory modalities, caution should be taken not to overgeneralize the present findings as broadly representative of sensory motor transformations (e.g., abstract, line 12).

Minor

1) Could a population (or even single cell) analysis be used to decode error vs correct responding prior to the first lick for error vs hit trials?

2) A visual vs auditory distinction for the role of PPC in decision making is presented in the introduction. Are the authors arguing that the present data suggests that mouse PPC does process auditory information, but only participates in decisions related to visual stimuli? It would be helpful to clarify this point, perhaps by revisiting this idea in the discussion.

3) p. 7. Line 139. Nontarget-preferring neurons in V1 are compared to target preferring V1 neurons using the engagement modulation index. Despite small numbers of nontarget-preferring neurons, could the same modulation index comparison be made for nontarget vs target-preferring neurons in PPC?

4) p. 7. Line 143 – It is not clear if task performance is synonymous with task engagement condition or both engagement and passive viewing conditions.

5) p. 7. Line 146 – Reference to figure 2?

6) Lines 182 and 188 - Hit and FA trials are compared while controlling for the possibility of motor confounds to see differences due to choice selectivity (line 188). However, why isn't the possibility of motor signal confounds addressed in CR vs FA trials (line 182)?

7) p. 11. Line 238. Why is "Engagement" a separate component for the Global GLM Analysis (Figure 4), but combined "stimulus + Engagement" here?

8) p. 11, line 236. How is R-squared of 0.995 significantly worse than 0.905?

9) p. 12. Line 269. Supplementary Figure 5 is referenced in the text before Supplementary Figure 4, the numbering for these figures should be reversed.

10) Line 590 - Mentions PMTs for first and only time without defining.

11) Line 75 and 636 – Some mice had both V1 and PPC imaged, according to the Statistics section (line 636). Earlier, authors state that imaging was for either/or not both regions (line 75). It is not clear at which time points the four mice had these areas imaged.

12) p. 38. Line 966 contains a typo "for each mouse using in imaging experiments."

13) Figure 3. Why does the peak for FA vs CR come significantly later than for hit vs FA for PPC?

Reviewer #3 (Remarks to the Author):

The manuscript entitled "Task-dependent representations of stimulus and choice in mouse parietal cortex" by Pho et al., examines the role of PPC in the mapping of sensations to actions in a visual decision making task. The authors conclude that PPC encodes sensory, decision, and motor variables, playing a central role in the sensory-motor transformation. This elegant work adds interesting pieces of information to the hotly debated and controversial topic on the role of mouse PPC in sensory-based decision making, and in my view it is particularly relevant for the understanding of how PPC is involved in motor planning (rather than execution).

I have two main criticisms. The first one can be briefly explained by referring to a statement in the discussion, Ln 440: "However, the possibility remains that the PPC responses recorded in our task reflect planning- or movement-related signals that originate elsewhere." Right after the inspection of Figure 1e, the reader assumes that planning- or movement-related signals converging onto PPC are the most likely explanation of the data. All subsequent pieces of evidence do not seem to contradict the simple view that stimulus-related and motor-planning related signals converge in PPC modulating the responses of neurons in this area. Please note my emphasis on motor planning and not on execution. However, the authors carefully avoid this interpretation, favoring instead vaguely-defined correlates of "choice" (as in the title) and "decision". I disagree with this solution, especially when referring to "decisions" (as in the Conclusion and outlook). My suggestion is to discuss the possibility of motor planning signals right away, spelling-out a working definition of "choice" that explicitly includes motor planning (or explaining the difficulty of separating the two in this work). Then conclude that mouse PPC carries (correlatively) an important motor-planning ("choice") component that, at least for the execution of motion, might not be causally relevant (as shown in their previous work, Goard et al., 2016). The arguments used to rule out a "purely motor" role of PPC (Ln 445-453) are certainly valid and in line with the idea of not pushing an extreme view of mouse PPC that would contradict a large body of literature on sensory-evidence modulation (accumulation).

This brings me to the second main criticism. When adopting this more natural framework it is unclear what the novelty of this study is relative to their previous work (Goard et al., 2016) where the authors concluded: "Population analyses demonstrated unique encoding of stimulus identity and behavioral choice information across regions, with V1 encoding stimulus, fMC encoding choice even early in the trial, and PPC multiplexing the two variables". The interesting hypothesis formulated in that study was that "PPC may be involved with converting stimulus identity representations into behavioral choice representations early in the task (during the stimulus epoch)". In my view the authors should try to nail down this point, the "transformational" element, but in its current version I struggled tremendously to find the relevant pieces of evidence throughout the manuscript.

When looking at the vast rodent literature on PPC, I personally think a number of observations made here are not trivial. The literature can be fairly confusing, highlighting different roles of PPC depending on the sensory modality (auditory, visual, tactile), and the timing relative to task components (during/after sensory-evidence accumulation). For example, as discussed by the authors correlative evidence has been provided that PPC neurons are linked to both evidence accumulation (Hanks et al., 2015) and choice (Raposo et al., 2014), with causal optogenetic experiments during stimulus presentation affecting also behavioral outcome (e.g. Licata et al., 2017 or work from the authors as well, Goard et al., 2016). Work by Alex Huk and colleagues (Katz et al., 2016) hinted also at the relevance of the spatial dissociation between stimulus location and RF location of the parietal neurons analyzed (although I believe such point cannot be addressed in this study given the large RF of neurons in mouse PPC and the stimuli used).

However, the role of PPC for motor planning (at least in the mouse) is less well understood. Based on

their previous work (Goard et al., 2016) the authors concluded: "it is unlikely that PPC is directly involved in executing motor plans, as we have previously shown that optogenetic inactivation of PPC during the response period, or even during the delay between stimulus and response, has no effect on behavior." A correlative analysis based on response variance (Licata et al., 2017) pointed to a different speculation for motor planning (not execution): "[...] the sharper decrease in the VarCE seen on visual decisions (Fig. 8E, blue) may reflect the dual stabilizing influences of visual sensory input and action planning feedback", indicating that PPC might carry (via feedback) a strong signature of the motor plan. Recent work from the Komiyama lab could have a similar interpretation (Hwang et al., 2017), although the pre-stimulus activity is there interpreted in the context of "choice biases" and not of (very early) motor planning. And a very similar work by Brody's lab has recently appeared in the bioRxiv (Akrami et al., 2017). Work in monkeys demonstrated instead a causal role of parietal cortices for movement execution (Hwang et al., 2012). I think a promising revised narrative that highlights the novelty of this work could be one that expands on the above considerations. The authors might agree with this consideration given their observation (Ln 290) that "Interestingly, some PPC cells encoded the stimulus in a contrast dependent manner, but also encoded the choice [i.e. motor planning] in a contrast-independent manner [...] in the same population of neurons".

Other comments:

When studying the stimulus-related selectivity, the authors used standard SDT methods sorting conditions with the same animal choice but different stimuli, i.e. hits vs FA (nicely equalizing the number of licks). However, unless I have completely misunderstood the task structure, they also "reverted" the reward value going from appetitive (water-hits) to aversive (quinine-FA). Similarly, for FA vs CR choice related selectivity (same stimuli) there is a strong aversive component in FA conditions. The authors should address the impact of value-change (water vs quinine) in these comparisons.

Still related to quinine, when was it delivered, immediately at the end of the stimulus presentation when the spout was made available? If so, could the animals have just used olfaction to solve the task, ignoring the visual stimulus? A simple control in trained animals could have been to skip quinine in a bunch of NT sessions (apologies if I missed it). Also, was quinine going through the very same "spout tube" as the water drops? If so, wouldn't it be almost impossible to avoid quinine contamination in water drops in Hit trials?

As the authors explain RL, A, and AM are visually responsive, retinotopically organized visual areas that can be (easily) identified in visual field sign maps, also in anesthetized animals. Furthermore, many neurons in these areas have strong orientation tuning (e.g. Fig. 6 in Marshel et al., Neuron 2011). When studying the stimulus-related selectivity (hits vs FA) part of the observed stimulus modulation could be due to a gross differences in the fraction of neurons preferring various stimulus orientations. Fig. 2e kind of addresses this issue (T, NT, i.e. horizontal and vertical). Why not showing the distribution of preferred orientations of the entire visually responsive population?

In the GLM "Binary predictors for licking indicated the duration of lick bouts, which were defined as groups of licks with an inter-lick interval less than 1 second". Since the authors seem to have an excellent control on the number of licks why not using licking frequency? Perhaps even outside the context of the GLM model, a strong correlation between licking frequency and response amplitude in target cells during engagement and stimulus presentation could provide further evidence of a significant pre-motor (motor planning) component.

In a few parts the authors make interesting observations followed by "data not shown". Please, show the data if relevant, and if not relevant please explain why. For example: Ln61 and 90 for not licking; L97 for cell type.

Please, add somewhere a brief explanation of what the animals were doing in miss trials: the spout advanced, water was made available, and the animal simply didn't lick? Are miss trials happening toward the end of a session for lack of motivation (not thirsty)? Or are they happening pseudo-randomly for a possible quinine contamination of water as mentioned before?

I could not find a description on how eye movements were monitored. I understand these are large stimuli, but I am more concerned that the animals might have been closing their eyes and used olfaction to solve the task, or simply took brief naps for a few trials, trials that were then labeled as misses. The common explanation: "we watched the videos and in general the animal was awake" sounds very approximate to me.

The motivational logic for the reversed contingency (Ln 329-333) is badly formulated: errors might be due to impulsivity so we reversed the motor contingency? Please rephrase.

The whole part on the reversed contingency is particularly "displeasing" to read outside the straightforward context of pre-motor/motor planning mentioned above. For example (Ln 347) "many PPC neurons did not exhibit stable stimulus selectivity, but instead appeared to track the animal's choice. These neurons were initially selective to target stimulus A and became selective to the new target stimulus B after reversal (Figure 7d)", this is almost a trivial observation when simply assuming those cells are linked to licking (e.g. lick preparation). Similarly (Ln 364-367) the whole quadrant analysis (Fig. 7), isn't it simply conveying the very same message that V1 follows the stimulus drive and PPC has instead lots of cells that correlate with licking and/or licking preparation? As mentioned above, the risk I see of pushing the line on choice and decision with no obvious evidence against a simpler interpretation (licking and/or licking preparation) is to irritate the reader, badly disposing him/her toward the many interesting observations in this work. If instead you are convinced you have strong evidence against such simpler interpretation (choice and possibly decisions vs motor planning), please present such evidence in a more organized and explicit way; overall I could only grasp some weak and unconvincing arguments against the motor-planning interpretation.

Results, p20. "mice withheld licking until the spout became available during the response epoch" – it is unclear how (and if) this was analyzed on all trials even considering Figure 3b. The infrared lick detector was attached to the spout according to Methods, so it seems it was impossible to detect attempts at licking. Also, in principle the mouse doesn't have to overtly lick – snout movements and whisking in anticipation of a spout would be enough for a neural response. How are the authors controlling for this?

Ln 372, "PPC truly contains..." please rephrase

Ln 397, what "behavioral state" are the authors referring to?

Ln 413, I think is anterior V1 and posterior somatosensory

Fig 6c, please better explain the top row. Are the responses in the "modulated by both" for contrast modulated only in the period during the stimulus presentation, but in the "modulated by contrast" also in the period after stimulus presentation? If so, the authors should address this difference (apologies if missed).

Methods, GLM: p16. Please use different variable names for R's otherwise equations are hard to interpret.

Discussion, P37, "Although some PPC neurons..." – wrong reference to a figure.

P38, "Second, we find a subset of neurons..." – wrong reference to a figure.

P38, "Finally, the responses of many PPC neurons..." – wrong figure reference.

Point-by-Point Response

We thank the reviewers for their time and effort in reviewing our manuscript. We were pleased to find that each of the reviewers recognized the significance and interest of our findings for the field. Their comments and critiques are thoughtful and much appreciated, and have served to improve the manuscript. In this revision, we have made several revisions to the text to address the reviewers' principal concerns and added supplementary analyses, while incorporating numerous additional minor changes as suggested by the reviewers.

The first major concern revolved around clear interpretation of the results given the nature of the go/no-go task. According to the suggestion of Reviewer 3, we have added a clear working definition of "choice" near the beginning of the results. "Choice" can include either decision formation or motor planning – as our task cannot distinguish between the two. We also have added caveats in interpreting the error trial analyses and clarified the motivation and interpretation of the reversal experiments.

The second major concern was the novelty of our study in relation to the rest of the field. We have clarified in our Introduction and Conclusion the current debate about rodent PPC, highlighting the novelty of our findings in relation to visual decisions, which add significant evidence that PPC is involved in sensorimotor transformation and not just retaining information about trial history or bias.

Finally, we addressed numerous technical concerns about the work. In particular, we have added supplemental analyses demonstrating that quinine does not affect future behavioral responses and analyzed the orientation tuning of PPC neurons for the Reviewers' interest. Additional technical concerns are addressed directly below.

Reviewer #1: Pho et al studied the function of PPC in a visual go-no go task and found that PPC appears to encode a mix of sensory and motor features. The authors used a combination of approaches including the analysis of error trials, modulation of the sensory stimulus, and reversals to aid understanding what is encoded in PPC. The biggest weakness of the paper is the task design, in which many behavioral variables are intermixed, like task engagement/attention and motor response (that is, the mouse only has to pay attention and create a motor response following the 'go'-related cue). I do not see any way for the authors to fully overcome this shortcoming. The use of error trials and miss trials are helpful but often it is hard to interpret error trials because it is unclear why the animal got the trial incorrect. I worry a bit that some behavioral differences between engaged/not engaged or contrast conditions could underlie the encoding the authors see in PPC. The authors have tried to control for things like licking, but I really do not see any way to eliminate or fully control for potential behavioral confounds given the go-no go task design. I do not have any major suggestions for improvement because the authors have done the best they can with the data they have and it would be unreasonable to ask for data from a better-designed behavioral task. Therefore, I am confident that the authors have done a reasonable job to try to overcome the limitations that arise from their task design.

We thank the reviewer for acknowledging the sufficiency of our analytical approaches for this paper. While we agree that we cannot *fully* control for all potential behavioral confounds with our go/no-go task design, we would argue that our simple basic task enabled us to add several

layers of complexity to the experiments. This, along with careful analysis of the data, has enabled us to address specific controversies regarding PPC and clarify its function (see below).

The results of PPC as a sensory and motor area likely will not greatly change how the field thinks about PPC. For me, those results confirm the majority view in the field. However, there is a minority of people who think of PPC as an area for visual representations, so perhaps the paper will be beneficial for that audience. From the authors' introduction, the main debate in the field appears to be whether PPC is a sensory-to-motor area for senses other than vision, which is not addressed here. Despite the limitations of the interpretation that is possible from the task design and despite the lack of an entirely novel result, the findings, in my opinion, will be useful for the field.

We thank the reviewer for pointing out that that our work will be useful for the field. Although the role of PPC for non-visual tasks is an interesting (and controversial) issue, we agree with the reviewer that we do not directly address this question in our paper. We have revised our introduction to focus on visual and visuomotor tasks (lines 20-36).

However, we would like to emphasize that there is considerable debate over the role of PPC in sensorimotor tasks, and a number of researchers would not find our results self-evident. We believe our findings go beyond simply the demonstration of PPC as a sensory and motor area. Based on our go-no go task, we carried out three experiments of increasing complexity - in which we examined the influence of engagement, contrast, and contingency reversal on PPC and V1 neurons. At the broadest level, we draw three important conclusions. First, we took advantage of manipulations to engagement and contrast to show that many individual PPC neurons multiplex visual and choice signals, with heterogeneous subpopulations showing differing modulation with stimulus contrast versus task engagement (Fig 6a-c). Second, we present evidence that PPC neurons exhibit engagement responses independently of future motor action (Fig. 4f), though we agree that all not all task confounds can be ruled out. Finally, we demonstrated that the vast majority of PPC neurons switch their responses following contingency reversal (Figure 7h), showing that although many PPC neurons have sensory-driven responses, these responses are remodeled depending on task demands. These findings further implicate PPC as a site of sensory-motor transformation, despite recent evidence that has disputed this conclusion.

Reviewer #2: Pho et al imaged neurons in mouse PPC and V1 during a go/no-go visual discrimination task to test the hypothesis that mouse PPC is critical for visual-motor transformations. The authors suggest that this modulation gives rise to flexible and heterogenous neuronal activity within the PPC that is distinct from activity modulation in V1, which seems to be primarily driven by visual input. The use of a paradigm where visual stimuli and the motor response are close together in time, and the temporal resolution of imaging makes it very difficult to disentangle planning and motor signals; however, the authors used a variety of manipulations to provide evidence for a PPC signal that does not appear to be purely sensory (particularly in contrast to V1 responses during the same task), nor purely motor. While it would be ideal to provide further evidence that the observed activity is not motor related (e.g., temporal profile centered on the initial motor response showing activity following sensory related

activity but preceding the actual motor response), such analyses are beyond the scope of the present paper that mostly acknowledges the boundaries of the evidence presented here.

We thank the reviewer for a nice summary of our work. The reviewer comments that further evidence would be ideal to demonstrate that PPC activity is not motor related. If we understand the comment correctly, the evidence the reviewer is looking for can be found in our prior manuscript (Goard et al, eLife 2016), which shows that a subset of PPC cells exhibit activity in the intervening delay period between stimulus and motor response. Furthermore, in the delay task the choice coding was highest shortly after stimulus presentation and became weaker over the course of the trial. In the revision, we note this additional evidence in our Discussion (Lines 469-472).

Major:

1) Some visual representation of the anatomical location of the data presented here would be very helpful, particularly given that the first point in the discussion is related to the anatomy of this region of the mouse parietal cortex. For example, could the data shown in Figure 6e be shown on a top down view of the mouse dorsal cortex?

The precise anatomical location of these imaging fields in relation to visual cortical areal boundaries (e.g. Garrett et al 2015) was not recorded for this dataset. However, the location of PPC in this series of experiments (cf. Goard et al., 2016) is now shown as an inset of Fig 6e. In separate retinotopic mapping experiments, we have shown that our imaged region overlaps with area AM (Hu, Rikhye, Goard, Sur, SfN abstract, 2016).

2) Why isn't the population number (e.g., median r-squared) reported for the stimulus+engagement model for PPC (but is for V1)? Wouldn't this be a critical factor for the argument that PPC is not motor or sensory?

The results for the stimulus+engagement model for PPC were depicted in Supp Fig 3, but we had left out the median r-squared from the text for brevity. However, the reviewer makes a good point, and we have revised the text to include this number. We found that the stimulus+engagement model performs worse than the full model for PPC (rel $R^2 = 0.91$ and 34% of cells with significantly worse fit), suggesting that motor signals are combined with stimulus and engagement signals in PPC.

3) Evidence for transformation of sensory information, other than visual information, into motor commands is not presented in this paper. Particularly given the current state of the literature suggesting differences with other sensory modalities, caution should be taken not to overgeneralize the present findings as broadly representative of sensory motor transformations (e.g., abstract, line 12).

We apologize for the confusion in our previous manuscript. We have made all conclusions specific to visual information. The abstract (line 12) has been changed to 'visual inputs'. Similarly, the end of the introduction (line 46), and the conclusions (lines 509 and 513) have been changed to say 'visual' or 'visuomotor'.

Minor

- 1) Could a population (or even single cell) analysis be used to decode error vs correct responding prior to the first lick for error vs hit trials? *We may be misunderstanding the reviewer's question, but we believe that this data is already present in Figure 3c-e for Hit vs FA, and Supp Fig 2c-e for Hit vs Miss. Decoding is performed at a single cell level using ideal observer analysis. For brevity we have not included population decoding in this manuscript, but see Goard et al 2016, Figure 6 for decoding of PPC responses in a related task.*
- 2) A visual vs auditory distinction for the role of PPC in decision making is presented in the introduction. Are the authors arguing that the present data suggests that mouse PPC does process auditory information, but only participates in decisions related to visual stimuli? It would be helpful to clarify this point, perhaps by revisiting this idea in the discussion. *We do not make any claims about PPC's role in processing auditory information, as our data does not involve auditory stimuli. As the multisensory nature of PPC is not a central point of our paper, we have revised the introduction to focus on visual and visuomotor functions of PPC (lines 20-36).*
- 3) p. 7. Line 139. Nontarget-preferring neurons in V1 are compared to target preferring V1 neurons using the engagement modulation index. Despite small numbers of nontarget-preferring neurons, could the same modulation index comparison be made for nontarget vs target-preferring neurons in PPC? *We excluded this for brevity. For the reviewer's interest, NT-preferring neurons exhibited a modulation index of 0.087 +/- 0.016 (n = 232 cells), whereas T-preferring neurons exhibited a modulation index of 0.351 +/- 0.004 (n = 3292 cells).*
- 4) p. 7. Line 143 – It is not clear if task performance is synonymous with task engagement condition or both engagement and passive viewing conditions. *Changed to "task engagement" for clarity.*
- 5) p. 7. Line 146 – Reference to figure 2? *Added.*
- 6) Lines 182 and 188 - Hit and FA trials are compared while controlling for the possibility of motor confounds to see differences due to choice selectivity (line 188). However, why isn't the possibility of motor signal confounds addressed in CR vs FA trials (line 182)? *In CR vs FA, we are explicitly looking for "choice" selectivity, which is broadly defined as any motor-, decision-, or action-related signal. We have clarified what we mean by "choice" in the text (Lines 162-164, 716-721, legend of Supp Table 1).*
- 7) p. 11. Line 238. Why is "Engagement" a separate component for the Global GLM Analysis (Figure 4), but combined "stimulus + Engagement" here? *The engagement component was designed to model differences in the stimulus response between Engaged and Passive conditions. A model that had an engagement component but no stimulus component is not easily interpretable – it would essentially capture stimulus responses on Engaged trials while modelling responses as inactive on all Passive trials. The stimulus+engagement model, by contrast, can be easily interpreted as the full model without any Motor component.*
- 8) p. 11, line 236. How is R-squared of 0.995 significantly worse than 0.905? *Thank you for catching our error – 0.995 is for stimulus+engagement; motor has been corrected to 0.156.*

9) p. 12. Line 269. Supplementary Figure 5 is referenced in the text before Supplementary Figure 4, the numbering for these figures should be reversed. Corrected.

10) Line 590 - Mentions PMTs for first and only time without defining. Replaced with "photomultiplier tubes".

11) Line 75 and 636 – Some mice had both V1 and PPC imaged, according to the Statistics section (line 636). Earlier, authors state that imaging was for either/or not both regions (line 75). It is not clear at which time points the four mice had these areas imaged. We interleaved sessions between areas, this is noted explicitly in the Statistics section.

12) p. 38. Line 966 contains a typo "for each mouse using in imaging experiments." Corrected, thank you.

13) Figure 3. Why does the peak for FA vs CR come significantly later than for hit vs FA for PPC? One reason may be that PPC responses encode both stimulus-related (FA vs CR) and choice-related (Hit vs FA) signals, which appear to follow different dynamics. This is consistent with a visuomotor transformation in which visual information is later transformed into choice.

Reviewer #3: The manuscript entitled "Task-dependent representations of stimulus and choice in mouse parietal cortex" by Pho et al., examines the role of PPC in the mapping of sensations to actions in a visual decision making task. The authors conclude that PPC encodes sensory, decision, and motor variables, playing a central role in the sensory-motor transformation. This elegant work adds interesting pieces of information to the hotly debated and controversial topic on the role of mouse PPC in sensory-based decision making, and in my view it is particularly relevant for the understanding of how PPC is involved in motor planning (rather than execution).

We thank the reviewer for acknowledging the relevance of our work for the PPC field.

I have two main criticisms. The first one can be briefly explained by referring to a statement in the discussion, Ln 440: "However, the possibility remains that the PPC responses recorded in our task reflect planning- or movement-related signals that originate elsewhere." Right after the inspection of Figure 1e, the reader assumes that planning- or movement-related signals converging onto PPC are the most likely explanation of the data. All subsequent pieces of evidence do not seem to contradict the simple view that stimulus-related and motor-planning related signals converge in PPC modulating the responses of neurons in this area. Please note my emphasis on motor planning and not on execution. However, the authors carefully avoid this interpretation, favoring instead vaguely-defined correlates of "choice" (as in the title) and "decision". I disagree with this solution, especially when referring to "decisions" (as in the Conclusion and outlook). My suggestion is to discuss the possibility of motor planning signals right away, spelling-out a working definition of "choice" that explicitly includes motor planning (or explaining the difficulty of separating the two in this work). Then conclude that mouse PPC carries (correlatively) an important motor-planning ("choice") component that, at least for the execution of motion, might not be causally relevant (as shown in their previous work, Goard et al., 2016). The arguments used to rule out a "purely motor" role of PPC (Ln 445-453) are certainly valid and in line with the idea of not pushing an extreme view of mouse PPC that would contradict a large body of literature on sensory-evidence modulation (accumulation).

We thank the reviewer for such a careful reading of our work, and for the useful suggestions. As the reviewer suggested, we have now spelled out our definition of “choice”, a source of confusion for other reviewers as well (Lines 162-164, 716-721, legend of Supp Table 1). As we describe in the text, we use the term choice since the signals could reflect motor planning, but may also reflect decision-related variables (e.g., sensory evidence, action selection). We have been careful to directly lay out these possibilities to avoid ambiguity. Indeed, in our previous work (Goard et al., 2016), we find a dependence on PPC in the stimulus period, but not during the delay period, suggesting that PPC activity may be necessary for initial action selection based on sensory evidence, serving to initiate sustained motor planning signals in frontal motor regions.

This brings me to the second main criticism. When adopting this more natural framework it is unclear what the novelty of this study is relative to their previous work (Goard et al., 2016) where the authors concluded: “Population analyses demonstrated unique encoding of stimulus identity and behavioral choice information across regions, with V1 encoding stimulus, fMC encoding choice even early in the trial, and PPC multiplexing the two variables”. The interesting hypothesis formulated in that study was that “PPC may be involved with converting stimulus identity representations into behavioral choice representations early in the task (during the stimulus epoch)”. In my view the authors should try to nail down this point, the “transformational” element, but in its current version I struggled tremendously to find the relevant pieces of evidence throughout the manuscript.

This paper adds a number of novel findings to our previous work. First, we show that PPC neurons are actively gated by engagement in a sensorimotor task. Second, we used analysis of variable contrast stimuli and a GLM model to further delineate the relative contributions of stimulus, engagement, and motor action to the responses of both V1 and PPC. These analyses showed that visual and motor signals are multiplexed *within individual neurons*. Finally, we showed that reversing reward contingencies causes the response selectivity to swap for the vast majority of PPC cells, indicating that the multiplexing of stimulus and response can be flexibly mapped depending on task contingencies. Taken together, these results bolster the evidence that PPC neurons are capable of mediating visuomotor transformations (although we agree that it does not constitute definitive proof). We have changed the discussion to clarify these points (lines 500-514).

When looking at the vast rodent literature on PPC, I personally think a number of observations made here are not trivial. The literature can be fairly confusing, highlighting different roles of PPC depending on the sensory modality (auditory, visual, tactile), and the timing relative to task components (during/after sensory-evidence accumulation). For example, as discussed by the authors correlative evidence has been provided that PPC neurons are linked to both evidence accumulation (Hanks et al., 2015) and choice (Raposo et al., 2014), with causal optogenetic experiments during stimulus presentation affecting also behavioral outcome (e.g. Licata et al., 2017 or work from the authors as well, Goard et al., 2016). Work by Alex Huk and colleagues (Katz et al., 2016) hinted also at the relevance of the spatial dissociation between stimulus location and RF location of the parietal neurons analyzed (although I believe such point cannot be addressed in this study given the large RF of neurons in mouse PPC and the stimuli used).

However, the role of PPC for motor planning (at least in the mouse) is less well understood. Based on their previous work (Goard et al., 2016) the authors concluded: “it is unlikely that PPC is directly involved in executing motor plans, as we have previously shown that optogenetic inactivation of PPC during the response period, or even during the delay between stimulus and response, has no effect on behavior.” A correlative analysis based on response variance (Licata et al., 2017) pointed to a different speculation for motor planning (not execution): “[...] the sharper decrease in the VarCE seen on visual decisions (Fig. 8E, blue) may reflect the dual stabilizing influences of visual sensory input and action planning feedback”, indicating that PPC might carry (via feedback) a strong signature of the motor plan. Recent work from the Komiyama lab could have a similar interpretation (Hwang et al., 2017), although the pre-stimulus activity is there interpreted in the context of “choice biases” and not of (very early) motor planning. And a very similar work by Brody’s lab has recently appeared in the bioRxiv (Akrami et al., 2017). Work in monkeys demonstrated instead a causal role of parietal cortices for movement execution (Hwang et al., 2012). I think a promising revised narrative that highlights the novelty of this work could be one that expands on the above considerations. The authors might agree with this consideration given their observation (In 290) that “Interestingly, some PPC cells encoded the stimulus in a contrast dependent manner, but also encoded the choice [i.e. motor planning] in a contrast-independent manner [...] in the same population of neurons”.

We are impressed and thankful for both the sophisticated analysis of the literature as well as the helpful suggestions of how to position our work. We have expanded our discussion (Lines 491-497) to highlight the relation of these studies to our work.

Other comments:

When studying the stimulus-related selectivity, the authors used standard SDT methods sorting conditions with the same animal choice but different stimuli, i.e. hits vs FA (nicely equalizing the number of licks). However, unless I have completely misunderstood the task structure, they also “reverted” the reward value going from appetitive (water-hits) to aversive (quinine-FA). Similarly, for FA vs CR choice related selectivity (same stimuli) there is a strong aversive component in FA conditions. The authors should address the impact of value-change (water vs quinine) in these comparisons.

The reviewer makes a good point. However, value differences would only affect selectivity for time points following delivery of reward, but not the stimulus-period selectivity that we focus on here. We have revised the text to clarify this point (Lines 716-721, legend of Supp Table 1).

Still related to quinine, when was it delivered, immediately at the end of the stimulus presentation when the spout was made available? If so, could the animals have just used olfaction to solve the task, ignoring the visual stimulus? A simple control in trained animals could have been to skip quinine in a bunch of NT sessions (apologies if I missed it). Also, was quinine going through the very same “spout tube” as the water drops? If so, wouldn’t it be almost impossible to avoid quinine contamination in water drops in Hit trials?

Quinine and water were delivered only after the first lick, eliminating the possibility of an olfactory strategy to solve the task. Quinine and water were delivered via two parallel tubes on the same lick spout. Contamination is impossible to rule out completely, but we believe it is negligible given the larger size of the water droplet and that any previous quinine would be consumed on the previous trial.

To further address the reviewer's concern, we analyzed the lick probability on trials following quinine consumption (i.e. following False Alarm trials). We found no significant decrease in licking following FA trials compared to other trials (see new **Supplementary Figure 1**) which argues that any possible quinine contamination was likely too weak to affect behavior.

As the authors explain RL, A, and AM are visually responsive, retinotopically organized visual areas that can be (easily) identified in visual field sign maps, also in anesthetized animals. Furthermore, many neurons in these areas have strong orientation tuning (e.g. Fig. 6 in Marshel et al., Neuron 2011). When studying the stimulus-related selectivity (hits vs FA) part of the observed stimulus modulation could be due to a gross differences in the fraction of neurons preferring various stimulus orientations. Fig. 2e kind of addresses this issue (T, NT, i.e. horizontal and vertical). Why not showing the distribution of preferred orientations of the entire visually responsive population?

To keep the task simple, we only used one target orientation and one non-target orientation. In a subset of mice, we did assess (passive) orientation tuning after training sessions were complete. Only a small fraction of PPC neurons exhibited significant Passive visual responses (5.2% versus 53.3% in V1).

We did not find any gross differences in the distribution of passive orientation preferences. This data has been included in **Response Figure 2** below for the reviewers' interest.

Response Figure 2: V1 and PPC have similar distribution of preferred orientations during passive visual stimulation.

The orientation tuning properties of a subset of cells was assessed using passive presentation of drifting grating stimuli. Cells with significant ($p < 0.05$, Bonferroni corrected) responses were split based on orientation selectivity index (OSI) into untuned ($OSI < 0.3$) and tuned ($OSI > 0.3$). While V1 has a much higher overall fraction of visually-responsive cells than PPC (note different y-scale), the distribution of preferred orientations among tuned cells was similar across V1 and PPC. For reference, orientations close to the target stimulus are in red, orientations close to the non-target are in blue.

In the GLM “Binary predictors for licking indicated the duration of lick bouts, which were defined as groups of licks with an inter-lick interval less than 1 second”. Since the authors seem to have an excellent control on the number of licks why not using licking frequency? Perhaps even outside the context of the GLM model, a strong correlation between licking frequency and response amplitude in target cells during engagement and stimulus presentation could provide further evidence of a significant pre-motor (motor planning) component.

We found that animals licked with a very stereotyped frequency (~7 Hz, see also Pinto & Dan 2015), which did not vary much between conditions or individual trials.

In a few parts the authors make interesting observations followed by “data not shown”. Please, show the data if relevant, and if not relevant please explain why. For example: Ln61 and 90 for not licking; L97 for cell type.

Ln 61 and 90: A consistent observation from our video recordings is that mice did not lick during passive blocks or during stimuli. This is supported by video analysis from a delayed response version of the task, in which no change in licking or facial movements were exhibited in Hit trials compared to Correct Reject trials (Figure 1 – Figure Supplement 1, Goard et al., 2016). Unfortunately, changes in luminance from the monitor during stimulus presentation prevented quantitative video analysis for the non-delayed version of the task used in this paper, so we have replaced “data not shown” with a reference to the relevant data (Line 62).

Ln 97: The quantification of cell types recorded with our imaging approach has already been published. We have replaced “data not shown” with a reference to our prior work (Line 98).

Please, add somewhere a brief explanation of what the animals were doing in miss trials: the spout advanced, water was made available, and the animal simply didn't lick? Are miss trials happening toward the end of a session for lack of motivation (not thirsty)? Or are they happening pseudo-randomly for a possible quinine contamination of water as mentioned before?

On Miss trials, the spout advances, and the animals choose not to lick. Water is not made available until after the animal would have licked. All consecutive Miss trials that occurred at the end of a session were not included in analysis (this has been clarified in the Methods, Lines 710-711).

While Response Figure 1 does show that a higher Miss rate following Non-target trials (FA or CR) compared to following Hit trials, the presence of quinine on prior trials had no effect (Miss rate following FA vs CR trials were similar).

I could not find a description on how eye movements were monitored. I understand these are large stimuli, but I am more concerned that the animals might have been closing their eyes and used olfaction to solve the task, or simply took brief naps for a few trials, trials that were then

labeled as misses. The common explanation: “we watched the videos and in general the animal was awake” sounds very approximate to me.

Unfortunately, we did not monitor eye movements for this dataset. As discussed above, the mice cannot solve the task using an olfactory strategy. Although we did not collect video for all mice, for our existing videos and in-person observations, we never observed mice closing their eyes for prolonged periods during behavior. As mentioned earlier, successive miss trials were very rare until after satiation, and these trials were removed from all analyses. Others in the lab have monitored eye movements during passive visual stimulation and found that well-habituated mice rarely, if ever, closed their eyes while head-fixed under the microscope (Breton-Provencher & Sur, SfN 2017).

The motivational logic for the reversed contingency (Ln 329-333) is badly formulated: errors might be due to impulsivity so we reversed the motor contingency? Please rephrase.

The reversal experiments are an important validation of our main hypothesis, that PPC reflects choice (decision or motor planning). The error analyses corroborate this hypothesis but remain inconclusive because errors can be difficult to interpret. We have improved the motivational logic to reflect this more clearly (lines 339-347).

The whole part on the reversed contingency is particularly “displeasing” to read outside the straightforward context of pre-motor/motor planning mentioned above. For example (Ln 347) “many PPC neurons did not exhibit stable stimulus selectivity, but instead appeared to track the animal’s choice. These neurons were initially selective to target stimulus A and became selective to the new target stimulus B after reversal (Figure 7d)”, this is almost a trivial observation when simply assuming those cells are linked to licking (e.g. lick preparation). Similarly (Ln 364-367) the whole quadrant analysis (Fig. 7), isn’t it simply conveying the very same message that V1 follows the stimulus drive and PPC has instead lots of cells that correlate with licking and/or licking preparation? As mentioned above, the risk I see of pushing the line on choice and decision with no obvious evidence against a simpler interpretation (licking and/or licking preparation) is to irritate the reader, badly disposing him/her toward the many interesting observations in this work. If instead you are convinced you have strong evidence against such simpler interpretation (choice and possibly decisions vs motor planning), please present such evidence in a more organized and explicit way; overall I could only grasp some weak and unconvincing arguments against the motor-planning interpretation.

The reviewer makes a good point (related to their earlier point above). For the reasons discussed above, we prefer the more inclusive term “choice” (including both motor and decision – related responses), particularly since many would not consider the PPC signal to encode a true “motor planning” signal. We have more specifically defined what we mean by choice and its relation to motor planning in the text (Lines 162-164, 716-721, legend of Supp Table 1).

We also agree that our original language describing PPC as having “altered stimulus selectivity” is confusing. We have re-written this section to fit with the Reviewer’s suggested interpretation, removing all language of “altered stimulus selectivity.” We have kept the word “selectivity” but clarified how to interpret this measure and our quadrant analysis. However, we respectfully disagree that the results of the experiment are trivial. All evidence prior to this point

in the manuscript indeed leads to the possibility of a choice-related signal, but do not conclusively demonstrate it the way the reversal experiments do.

Results, p20. “mice withheld licking until the spout became available during the response epoch” – it is unclear how (and if) this was analyzed on all trials even considering Figure 3b. The infrared lick detector was attached to the spout according to Methods, so it seems it was impossible to detect attempts at licking. Also, in principle the mouse doesn’t have to overtly lick – snout movements and whisking in anticipation of a spout would be enough for a neural response. How are the authors controlling for this?

We visually confirmed mice withheld licking but concede that we did not record or analyze every trial. Video analysis from a subset of delayed-response experiments confirmed that mice did not exhibit facial movements or lick even in the prolonged period before response (Figure 1 – figure supplement 1 in Goard et al., 2016)

Ln 372, “PPC truly contains...” please rephrase Rephrased to “If PPC includes distinct populations of “stimulus” neurons and “choice” neurons...”

Ln 397, what “behavioral state” are the authors referring to? Revised to “task engagement”.

Ln 413, I think is anterior V1 and posterior somatosensory We apologize for the confusing wording, we have revised appropriately.

Fig 6c, please better explain the top row. Are the responses in the “modulated by both” for contrast modulated only in the period during the stimulus presentation, but in the “modulated by contrast” also in the period after stimulus presentation? If so, the authors should address this difference (apologies if missed). All neurons are categorized based on their responses during the stimulus period. The time courses in the top row include post-stimulus data, but this is not used to generate the response amplitudes in the middle row. We have clarified the description in the legend.

Methods, GLM: p16. Please use different variable names for R’s otherwise equations are hard to interpret. We apologize, but we are not sure what is requested here. The GLM equation is on page 30, and we do not have any variables named R in that equation. We do refer to absolute and relative R², which are standard uses of this variable.

Discussion, P37, “Although some PPC neurons...” – wrong reference to a figure.

P38, “Second, we find a subset of neurons...” – wrong reference to a figure.

P38, “Finally, the responses of many PPC neurons...” – wrong figure reference.

Thank you; we corrected the first and third figure references (the second one was correct).

REVIEWERS' COMMENTS:

Reviewer #1 (Remarks to the Author):

The authors have fully addressed all my comments. I think this paper will be a valuable addition to the field.

Reviewer #2 (Remarks to the Author):

Pho et al imaged neurons in mouse PPC and V1 during a go/no-go visual discrimination task to test the hypothesis that mouse PPC is critical for visual-motor transformations. As discussed in the original review of this paper, limitations of the approach make it difficult to fully disentangle planning and motor signals; however, the authors use a variety of manipulations to provide evidence for a PPC signal that does not appear to be purely sensory, nor purely motor.

The revised version of this manuscript addressed my previous concerns and I feel that this paper does advance the field in its current form.

Reviewer #3 (Remarks to the Author):

The authors have addressed my main concerns and criticisms. I have only one further suggestion: in this revised version the authors have introduced the terminology "multiplexing", observing that "a third group of neurons have multiplexed selectivity to stimulus, engagement, and motor signals, exhibiting complex responses to combinations of sensory or motor signals in engaged conditions". I agree with this choice, being very relevant for the stimulus-to-decision transformation. However the authors should help the reader accessing the key quantifications. What are the specific statistical analyses that define such third group? Bits and pieces of information are scattered throughout the manuscript, e.g. 20% of neurons are selective for A or B, but after reversal ~7% retain stim identity (?). Or, 40% of neurons have contrast modulation; and yet, "a small subset of PPC neurons (6.9% ± 3.7%) did show stable stimulus selectivity", etc. I'd suggest to add in the discussion a clear definition for this third group, the proportion of cells included, and a short summary of the related quantifications.

Minor

- Apologies if I misunderstood the caption, but Supp. Fig. 1 clearly shows that the hit rate decreases when quinine was delivered in the previous trials (previous FA), compared to water (previous hit). The main point seems to be that such decrease is the same for previous FA or CR, suggesting a "repeat-previous-behavior" strategy. I would make this clear and also report the related quantifications.

- "We define "choice" selectivity broadly as any signal related to behavioral output independent of the stimulus". This definition clearly includes motor execution. I would try to clarify (in the definition itself) that this is not the case.

REVIEWERS' COMMENTS:

Reviewer #1 (Remarks to the Author):

The authors have fully addressed all my comments. I think this paper will be a valuable addition to the field.

We are grateful for the reviewer's comments and the time spent in reviewing the manuscript.

Reviewer #2 (Remarks to the Author):

Pho et al imaged neurons in mouse PPC and V1 during a go/no-go visual discrimination task to test the hypothesis that mouse PPC is critical for visual-motor transformations. As discussed in the original review of this paper, limitations of the approach make it difficult to fully disentangle planning and motor signals; however, the authors use a variety of manipulations to provide evidence for a PPC signal that does not appear to be purely sensory, nor purely motor. The revised version of this manuscript addressed my previous concerns and I feel that this paper does advance the field in its current form.

We are grateful for the reviewer's comments and the time spent in reviewing the manuscript.

Reviewer #3 (Remarks to the Author):

The authors have addressed my main concerns and criticisms. I have only one further suggestion: in this revised version the authors have introduced the terminology "multiplexing", observing that "a third group of neurons have multiplexed selectivity to stimulus, engagement, and motor signals, exhibiting complex responses to combinations of sensory or motor signals in engaged conditions". I agree with this choice, being very relevant for the stimulus-to-decision transformation. However the authors should help the reader accessing the key quantifications. What are the specific statistical analyses that define such third group? Bits and pieces of information are scattered throughout the manuscript, e.g. 20% of neurons are selective for A or B, but after reversal ~7% retain stim identity (?). Or, 40% of neurons have contrast modulation; and yet, "a small subset of PPC neurons (6.9% ± 3.7%) did show stable stimulus selectivity", etc. I'd suggest to add in the discussion a clear definition for this third group, the proportion of cells included, and a short summary of the related quantifications.

We are grateful for the reviewer's comments and time spent in reviewing the manuscript.

The third multiplexed group was identified using the contrast task, where 28% of PPC neurons exhibited modulation by both stimulus contrast and task engagement. The partial GLM models (Supplementary Figure 4) also point to the multiplexed nature of these neurons, as 34% of cells had significantly worse fits for partial models compared to the full GLM model. As suggested by the reviewer, we have added these definitions with additional quantifications and Figure subpanel references to the Discussion (Lines 455-466).

Minor

- Apologies if I misunderstood the caption, but Supp. Fig. 1 clearly shows that the hit rate decreases when quinine was delivered in the previous trials (previous FA), compared to water (previous hit). The main point seems to be that such decrease is the same for previous FA or CR, suggesting a “repeat-previous-behavior” strategy. I would make this clear and also report the related quantifications.

We have revised the caption to more clearly emphasize our point, which is that consumption of quinine on FA trials does not reduce lick rate relative to CR. (The comment about strategy is interesting but not relevant to our point so we prefer not to speculate). We have also added the relevant statistical tests.

- “We define “choice” selectivity broadly as any signal related to behavioral output independent of the stimulus”. This definition clearly includes motor execution. I would try to clarify (in the definition itself) that this is not the case.

Technically our definition of choice selectivity does in fact include motor execution, but only for time points after the onset of the choice period. To clarify, we’ve updated our definition accordingly: “We define “choice” selectivity broadly as any premotor signal related to behavioral output independent of the stimulus”.